# HIBRID: histology-based risk-stratification with deep learning and ctDNA in colorectal cancer

Chiara M. L. Loeffler[1,2,3], Hideaki Bando [1][4,5,6], Srividhya Sainath[1], Hannah Sophie Muti[1,2,7], Xiaofeng Jiang [1][1], Marko van Treeck[1], Nic Gabriel Reitsam [1][1,8,9], Zunamys I. Carrero [1][1], Asier Rabasco Meneghetti [1][1], Tomomi Nishikawa[4], Toshihiro Misumi[4], Saori Mishima[5], Daisuke Kotani [1][5], Hiroya Taniguchi [1][10], Ichiro Takemasa[11], Takeshi Kato[12], Eiji Oki [1][13], Yuan Tanwei [1][14], Wankhede Durgesh[14], Sebastian Foersch[15], Hermann Brenner [1][14,16], Michael Hoffmeister [1][14], Yoshiaki Nakamura [1][5,6], Takayuki Yoshino [1][4,5,6] ✉ & Jakob Nikolas Kather [1][1,2,3,16] ✉

Although surgical resection is the standard therapy for stage II/III colorectal cancer, recurrence rates exceed 30%. Circulating tumor DNA (ctNDA) detects molecular residual disease (MRD), but lacks spatial and tumor microenvironment information. Here, we develop a deep learning (DL) model to predict disease-free survival from hematoxylin & eosin stained whole slide images in stage II-IV colorectal cancer. The model is trained on the DACHS cohort ($n = 1766$) and validated on the GALAXY cohort ($n = 1404$). In GALAXY, the DL model categorizes 304 patients as DL high-risk and 1100 as low-risk (HR 2.31; $p < 0.005$). Combining DL scores with MRD status improves prognostic stratification in both MRD-positive (HR 1.58; $p < 0.005$) and MRD-negative groups (HR 2.1; $p < 0.005$). Notably, MRD-negative patients predicted as DL high-risk benefit from adjuvant chemotherapy (HR 0.49; $p = 0.01$) vs. DL low-risk (HR = 0.92; $p = 0.64$). Combining ctDNA with DL-based histology analysis significantly improves risk stratification, with the potential to improve follow-up and personalized adjuvant therapy decisions.

Colorectal cancer (CRC) is one of the leading causes of cancer-related deaths worldwide[1]. Surgical resection remains the standard curative therapy in patients with stages II-III CRC and resectable metastases. Despite advancements in surgical and adjuvant therapies, recurrence rates remain substantial up to 30% for stages III CRC and up to 60% in resectable metastatic CRC[2–5]. Patients who relapse have an increased mortality risk, hence identifying these patients at an early stage is crucial for optimizing follow-up treatment decisions. Current prognostication systems for risk assessment, including imaging techniques, clinicopathological features and molecular data, such as *BRAF, RAS* mutational status and microsatellite instability (MSI), are moderate predictors for recurrence risk[6–8]. Similarly, follow-up strategies, such as tumor marker monitoring with carcinoembryonic antigen (CEA), lack sensitivity and specificity in identifying recurrence[9–11]. In particular for stage II CRC, the decision on adjuvant chemotherapy (ACT) is based on diverging risk assessment recommendations provided through international oncological associations[12,13]. Thus, a more fine-grained system for estimating the risk of relapse is required, as no stage-specific survival benefit for adjuvant chemotherapy has been proven. Therefore, new biomarkers for better and more precise prognostication are needed. Circulating tumor DNA (ctDNA) has emerged as a promising minimally invasive biomarker that measures a

**Fig. 1 | Study Design and DL Model Architecture. A** DACHS cohort and **B** GALAXY cohort overview including patient characteristics and WSI preprocessing pipeline using UNI, a pretrained vision encoder for feature extraction. **C** Overview Experimental Setup: Clinical data is fed into DL Model with WSIs for training process and then externally deployed onto the GALAXY cohort to obtain the DL-Score, which are then binarized into DL high-risk and DL low-risk categories. **D** Architecture of the Transformer-based Multiple Instance Learning (MIL) pipeline. WSIs are divided into patches and preprocessed to feature vectors with a dimension of n-tiles x1024 using the UNI foundation model. Patch feature vectors are then projected to a 512-

dimensional vector using a fully connected layer with ReLU activation, with a learnable class token (CLS) added. A two-layer transformer refines the CLS token via self-attention and feedforward networks. The final CLS token, encoding WSI-level information, is processed by an MLP to generate the patient-level risk score. This Figure was partly generated using Flaticon. DACHS=Darmkrebs: Chancen der Verhütung durch Screening Study, WSI=whole-slide image, DFS=disease-free survival, DL=Deep Learning, MRD=molecular residual disease, CLS=class learnable token, MLP=multilayer perceptron.

small fraction of cell free ctDNA in the blood, allowing for the detection of molecular residual disease (MRD) status[14,15]. Additionally, ctDNA can be used for monitoring treatment response and early prediction of recurrence, as ctDNA positivity after surgery is associated with a higher risk of disease recurrence[16,17]. Previous studies have shown that this correlation had already been found as early as four weeks after primary tumor resection[18]. However, ctDNA analysis alone does not capture the morphological characteristics of the tumor. For instance, histopathological features such as subtype, grading, vascular and lymphatic invasion, as well as the abundance of tumor-infiltrating lymphocytes[19–21] have been shown to be prognostically relevant and are reflected in clinical guidelines[13,22]. In addition, molecular features like MSI are included in treatment recommendations due to their association with prognosis[23,24]. Deep Learning (DL) is an artificial intelligence technology that is useful to extract quantitative biomarkers from routinely available clinical data in oncology[25,26]. DL models, trained on histopathological routine hematoxylin and eosin (H&E) tumor slides have been shown to act as survival prediction models augmenting current risk-stratifications systems[27–29]. DL can extract highly relevant information from routine pathology slides of

CRC, including presence of MSI[30,31], genetic alternations[31,32], response to neoadjuvant therapy[33], and overall survival (OS)[28]. Given the ability of DL to extract meaningful biological information from pathology slides that ctDNA cannot capture, we hypothesize that the combination of MRD assessment with a transformer-based DL risk score from morphology could significantly improve prognosis prediction.

Here, we show that combining MRD status from ctDNA with a DL-based risk score derived from routine WSIs significantly improves patient stratification and recurrence prediction in patients with CRC.

## Results
### DL stratifies patients by recurrence risk
We trained a DL model to generate risk scores based on DFS and validated its performance on the GALAXY cohort. Patients were categorized into DL high- and DL low-risk and their recurrence risk was analyzed. Among the 1404 patients 21.7% ($n = 304$) were categorized as DL high-risk and 78.3% ($n = 1100$) as DL low-risk (Fig. 1, Supplementary Fig. 1). Patients classified as DL high-risk exhibited a significantly elevated risk of disease recurrence compared to DL low-risk patients (HR = 2.31, CI 95% 1.86–2.86; $p < 0.005$), with a 24-month DFS of 57.6%

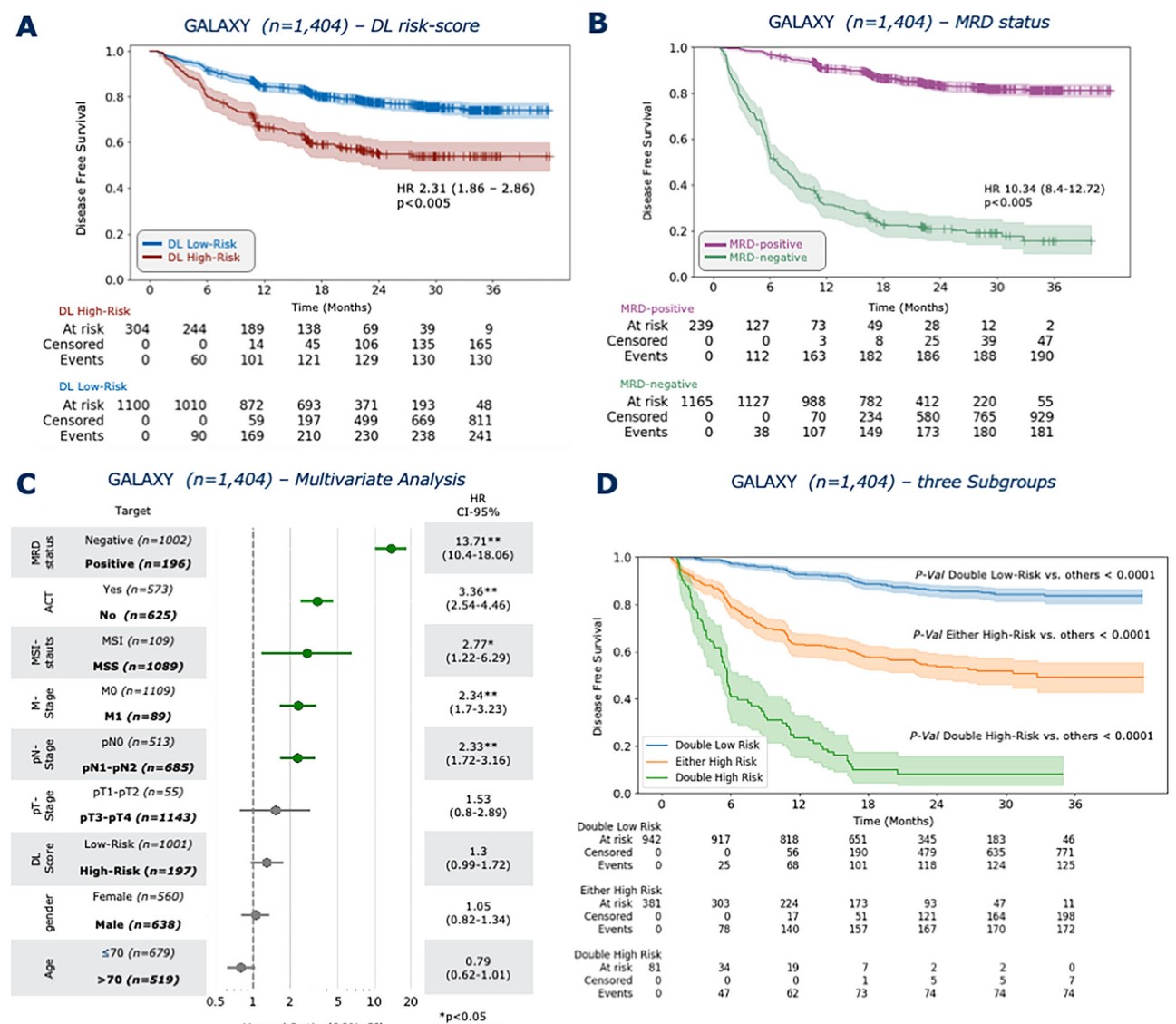

**Fig. 2 | DL and ctDNA-status stratify patients by recurrence risk. A** Kaplan-Meier curves for DFS stratified by DL high-risk and DL low-risk patients. **B** Kaplan-Meier curves for DFS stratified by MRD-positive and MRD-negative patients. **C** Forest plot showing multivariate cox regression analysis including the covariates gender ($p = 0.70$), age ($p = 0.06$), DL risk score ($p = 0.06$), pathological nodal stage (pN), pathological tumor stage (pT; $p = 0.20$), metastasis stage (pM), adjuvant chemotherapy treatment (ACT), microsatellite instability status (MSI; $p = 0.01$), and MRD-status, and their association with DFS. **D** Kaplan-Meier curves for DFS stratified into three distinct risk categories: Double High Risk (MRD-positive and DL high-risk), Either High Risk (either MRD-positive/DL low-risk or MRD-negative/DL high-risk), and Double Low Risk (MRD-negative/DL low-risk). HR and 95% CI were calculated by the Cox proportional hazard model. *P* value was calculated using the two-sided log-rank test (*$p < 0.05$, **$p < 0.005$). *P* values < 0.005 are not listed individually. Each Kaplan–Meier analysis was performed once using the full cohort and reflects the entire dataset. No subsampling or repeated trials were applied. Plots were generated using the lifelines package in Python 3.11.5. Source data is provided as a Source Data file. DACHS=Darmkrebs: Chancen der Verhütung durch Screening Study, WSI=whole-slide image, DFS=disease-free survival, DL=Deep Learning, MRD=molecular residual disease, HR=Hazard ratio, CI=Confidence interval.

vs. 79.1%, respectively (Fig. 2A). In the landmark analysis excluding early recurrences within 3 and 6 months, the hazard ratios (HRs) were 2.37 (1.88–3; $p < 0.0001$) and 2.14 (1.61–2.84; $p < 0.0001$) respectively, which confirmed consistent trends with the primary analysis. (Supplementary Fig. 2A, B). In the stage-specific analysis the DL risk score significantly stratified the stage III CRC patients with an HR of 1.87 (CI% 1.35–2.61, $p < 0.0005$), whereas for stage II and IV the DL score was not significant (Supplementary Fig. 3A–C). For comparison, ctDNA analysis alone stratified 17 % ($n = 239$) patients as MRD-positive and 83% ($n = 1165$) as MRD-negative, with an HR of 10.34 (CI 95% 8.4–12.72, $p < 0.005$, Fig. 2B). In the multivariate analysis, we found the most prognostic indicator for recurrence risk to be MRD positivity (HR = 13.71, CI 95% 10.4–18.06; $p < 0.005$), followed by adjuvant

chemotherapy treatment (HR = 3.36, CI 95% 2.54–4.46; $p < 0.005$, Fig. 2C). The DL-risk score was not significant with an HR of 1.3 (CI 95% 0.99–1.72, $p = 0.06$). When correlating the DL risk categories with patient characteristics of the validation cohort, we found significant differences in sex, pT-Stage, pN-Stage, pathological Stage, and MRD status (Supplementary Table 1). Together, these results demonstrate that our DL model can significantly stratify patients according to their risk of recurrence.

**DL stratifies recurrence risk within MRD subgroups**
We hypothesized that combining MRD status with our DL risk scores can further improve patients stratification four-weeks after curative surgery, particularly the MRD-negative patients. First, we grouped the

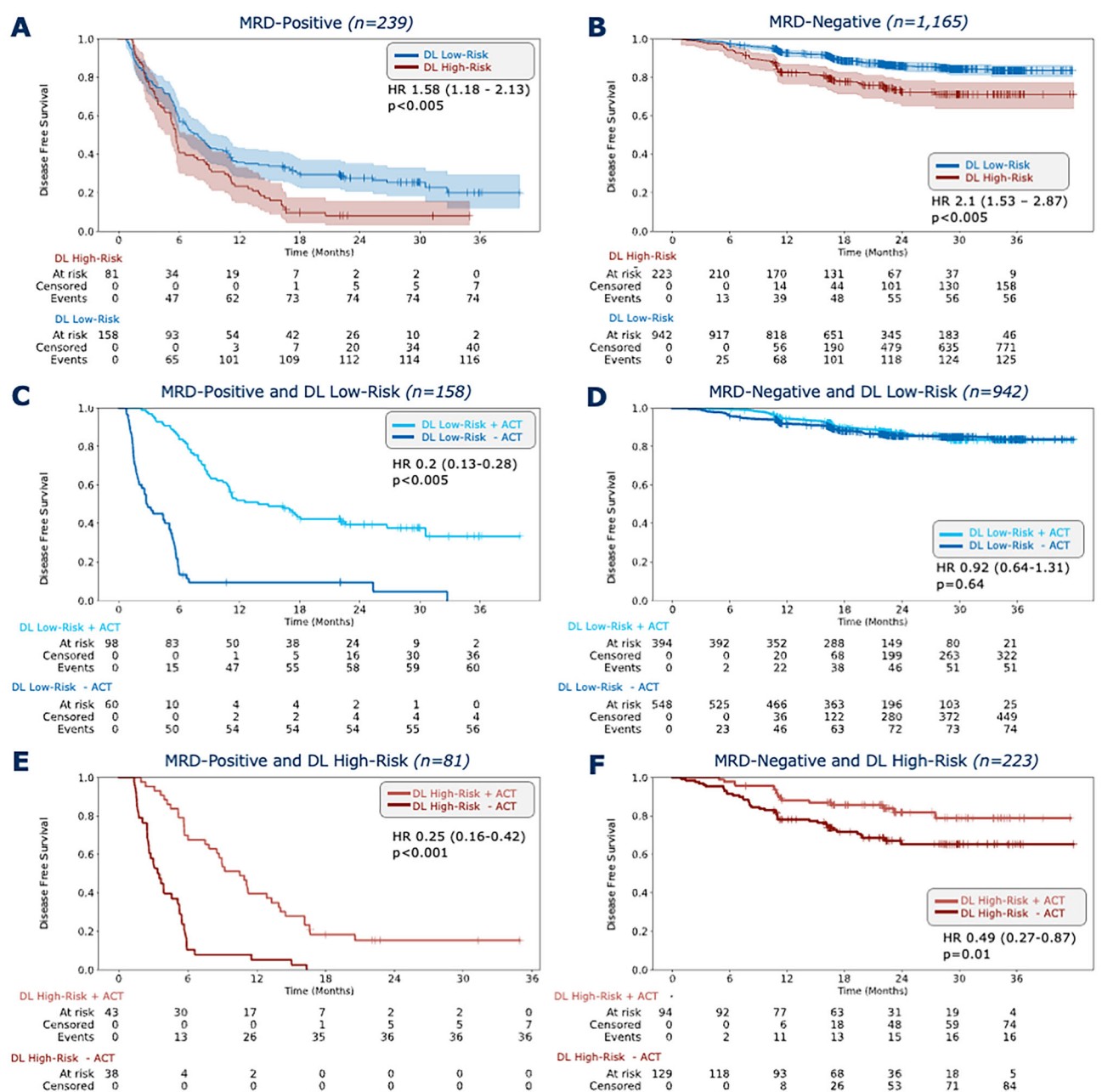

**Fig. 3 | DL stratifies recurrence risk within MRD subgroups.** Kaplan-Meier curves showing DFS stratification by DL high-risk and DL low-risk groups for **A** MRD-positive and **B** MRD-negative groups, followed by Kaplan-Meier curves showing DFS stratified by with or without ACT treatment in **C** MRD-positive and DL low-risk, **D** MRD-negative and DL low-risk, **E** MRD-positive and DL high-risk and **F** MRD-negative and DL high-risk subgroups. HR and 95% CI were calculated by the Cox proportional hazard model. *P* value was calculated using the two-sided log-rank test. Plots were generated using the lifelines package in Python 3.11.5 Source data is provided as a Source Data file. DFS=disease-free survival, DL=Deep Learning, ACT=adjuvant chemotherapy, MRD=molecular residual disease, HR=Hazard ratio, CI=Confidence interval.

patients into three distinct risk categories: Double High Risk (MRD-positive and DL high-risk), Either High Risk (either MRD-positive/DL low-risk or MRD-negative/DL high-risk), and Double Low Risk (MRD-negative/DL low-risk). These categories showed significant differences in DFS (Fig. 1D). In the MRD-positive group, 33.9% (81 out of 239 patients) were categorized as DL high-risk and 66.1% (158 out of 239 patients) as DL low-risk. Patients in the DL high-risk groups had significantly worse outcomes with an HR of 1.58 (CI 95% 1.18–2.13; $p < 0.005$, Fig. 3A). The DFS-time interval was longer in the DL low-risk group, with a 24-months DFS of 29.1% compared to 8.6% in the DL high-risk group (Fig. 3A). In the MRD-negative group, 19.1% (223 out of 1165 patients) were classified as high-risk by the DL model and 80.9% (942 out of 1165 patients)

as DL low-risk with an HR of 2.1 (CI 95% 1.53–2.87; $p < 0.005$, Fig. 3B). Additionally, the 24-month DFS was longer in the DL low-risk group at 87.4%, compared to 75.3% in the DL high-risk group (Fig. 3B). In a multivariate Cox analysis with age, gender, pT, pN and pM as covariates, the DL-score was the only independent prognostic predictor in the MRD-positive group with an HR of 1.51 (CI 95% 1.06–2.15; $p = 0.02$, Supplementary Fig. 2C). In the MRD-negative group, pM was the strongest prognostic indicator with an HR of 2.39 (CI 95% 1.43–4.01; $p < 0.001$, Supplementary Fig. 2D), while the DL risk score was not an independent prognostic predictor (HR = 1.08 CI 95% 0.68–1.72; $p = 0.75$, Supplementary Fig. 2D). In summary, these data show that the combination of MRD status with the DL risk score

provides improved patients stratification, with the DL risk score being independently prognostic only in the MRD-positive group.

### DL-based recurrence risk predicts benefit from adjuvant chemotherapy in MRD-negative patients

We hypothesized that our DL-risk score could identify patients with stage II-IV CRC who might benefit from ACT, despite being MRD-negative (Fig. 3 C-F). For the MRD-positive group, ACT significantly improved DFS in both the DL low-risk group (HR = 0.20, CI 95% 0.13–0.28; $p < 0.001$, Fig. 3C) and in the DL high-risk group (HR = 0.25, CI 95% 0.16–0.42; $p < 0.001$, Fig. 3E). Without receiving ACT, all MRD-positive and DL high-risk patients experienced recurrence within 24-months, whereas 16.3% of those receiving ACT remained disease-free after 24 months (Fig. 2E). In the MRD-negative group, patients in the DL low-risk group did not benefit from ACT (HR = 0.92, CI 95% 0.64–1.31; $p = 0.64$). The 24-month DFS was 88.3% for patients treated with ACT vs 86.7% for patients not receiving ACT (Fig. 3D). Interestingly, patients in the MRD-negative and DL high-risk group showed significantly longer DFS when treated with ACT (HR 0.49, CI 95% 0.27–0.87; $p = 0.01$, Fig. 3D). The 24-month DFS rate was 84% in patients who received ACT and thus significantly higher than in patients who did not receive ACT (69%). This disease-free survival advantage persisted at 36-months, with DFS rate of 83% (with ACT) vs. 69% (without receiving ACT, Fig. 2F). The DFS trends among different groups remained consistent in the 3-months performed landmark analysis (Supplementary Fig. 4). This indicated that even within the low-risk subgroup (according to MRD), there are high-risk individuals for whom the omission of ACT may carry a higher risk of recurrence. However, in a stage-specific analysis of MRD-negative and DL high-risk patients, compromising a smaller subset of individuals, no significant DFS difference was observed (Supplementary Fig. 3D–F). Patient characteristics between those receiving ACT and those who did not revealed significant differences in age, ECOG-status, pT, pN and MSI-status (Supplementary Table 2). Together, these data show that the DL prognostication model can successfully further stratify MRD-negative patients.

### DL as a tool for prognostic histopathological discovery

While measurements of ctDNA provides information about viable and disseminated tumor cells and enables minimal-invasive MRD assessment, it does not capture any information regarding tumor morphology or the TME, both of which are reflected in histopathology slides and known to impact clinical outcomes. We evaluated whether our DL model, trained without manual annotations, identifies morphological features of the tumor and TME synergistic to MRD status. Therefore, we visualized highly predictive regions at multiple magnifications in the GALAXY cohort. (Fig. 4). In the DL low-risk classified patients, the morphological analysis revealed a variety of benign histopathological tissue features. As the DL score increased, the histological image tiles still below the risk threshold displayed moderately differentiated tumor components. These samples still displayed a balanced tumor-stroma ratio and tumor glands with tubular to cribriform architecture, indicating an intermediate phenotype between DL low and DL high-risk morphological characteristics (Fig. 4A). The images, above the risk threshold, displayed high-grade tumor cells with a significant desmoplastic stroma reaction. There was a high intratumoral stroma fraction, and the presence of tumor buds/poorly differentiated clusters, which are known to be associated with a higher recurrence risk (Fig. 4B)[34–38]. Taken together, we observed a clear morphological continuum mirroring the progression from DL low to DL high-risk tumors. Moreover, we analyzed the distribution of the DL risk score with clinically relevant molecular information, namely MSI status, *BRAF* and *RAS* mutational status (Supplementary Fig. 5A–C). We found that the distribution was very similar for all these factors,

suggesting that our DL model independently detects and accounts for additional prognostically relevant morphological features.

## Discussion

CRC can often be cured through surgery; however, a subset of patients, particularly those with metastatic colorectal cancer (mCRC), experience relapse, associated with high mortality. To mitigate this risk, ACT is administered to mCRC patients post-surgery. However, not all of these patients benefit equally from such treatment, which is associated with substantial side effects[39–42]. Decades of research have focused on identifying potential biomarkers to administer ACT selectively to high-risk individuals who would benefit the most, while withholding it from low-risk individuals. To date, one of the most promising biomarkers for this purpose is ctDNA. Measurement of MRD through ctDNA is minimal-invasive, robust, and highly prognostic. However, ctDNA does not capture the tumor's interaction with its microenvironment—the complex spatial ecology of tumors[43]-nor the tumor morphology itself. This is a limitation of ctDNA as a biomarker, given that, in addition to conventional histopathology tumor features, the interplay between tumors and their microenvironment has been demonstrated to be highly prognostic and predictive over the years. In our study, we demonstrate that combining DL-based risk assessment with MRD measurement further enhances prognostic capabilities: MRD-negative patients who were predicted to be at high-risk for relapse by our DL model had a significantly longer DFS if treated with ACT, whereas in MRD-negative patients with a DL-based low-risk status no DFS benefit was seen for those receiving ACT (Fig. 3F). These observations suggest that healthcare providers may identify a subset of patients who are at risk for relapse but are not detected through current diagnostic tools, including a diagnostic as innovative as ctDNA. On the other hand, our model contributes to identifying patients who can safely forgo ACT to minimize the burden of unnecessary toxicity without compromising outcomes. This could support customized treatment decisions and reduce overtreatment of low-risk patients. However, we acknowledge that patients who did not receive ACT in our cohort more often had unfavorable baseline characteristics—such as older age, higher ECOG performance status, and a higher proportion of stage IV disease—factors that are independently linked to poorer outcomes. The pN-status demonstrated strong prognostic value in our study. While primary tumor histology contains features partially predictive of pN-status[27,44], combining explicit pN-status with DL model outputs, as shown by Jiang et al.[28] may further enhance prognostic performance. Recent studies[45] demonstrate the potential of integrating histological and clinical data for multimodal outcome prediction. However, further research is required to determine whether multimodal training or post-hoc integration yields better risk stratification. Previous studies developing DL-based prognostication systems failed to provide evidence for potentially different chemotherapy efficacy across DL categories, by which all potential therapeutic implications of these models remain speculative[29,46] Our study builds upon this foundation by incorporating an additional key finding: although our histopathology DL model was trained without human annotations and solely on raw WSIs, we found that the model learned to pay attention to regions linked to tumor biological features plausibly associated with prognosis, thereby synergizing with ctDNA. The DL model combines contributions from all tiles in a nonlinear manner - a process that is learned during the training. As a result, high- and low-risk tiles do not cancel each other out. The tile-level heatmaps provide interpretability by highlighting prognostically relevant regions, which could complement traditional pathology evaluation in diagnostic settings. However, heatmaps cannot visualize all nonlinear operations in the model and improved explainability techniques remain a subject of ongoing investigation[47]. Our findings are consistent with previous DL-based end-to-end prognostication approaches in CRC based on H&E histopathology alone[28,48]. To our knowledge, our study provides initial

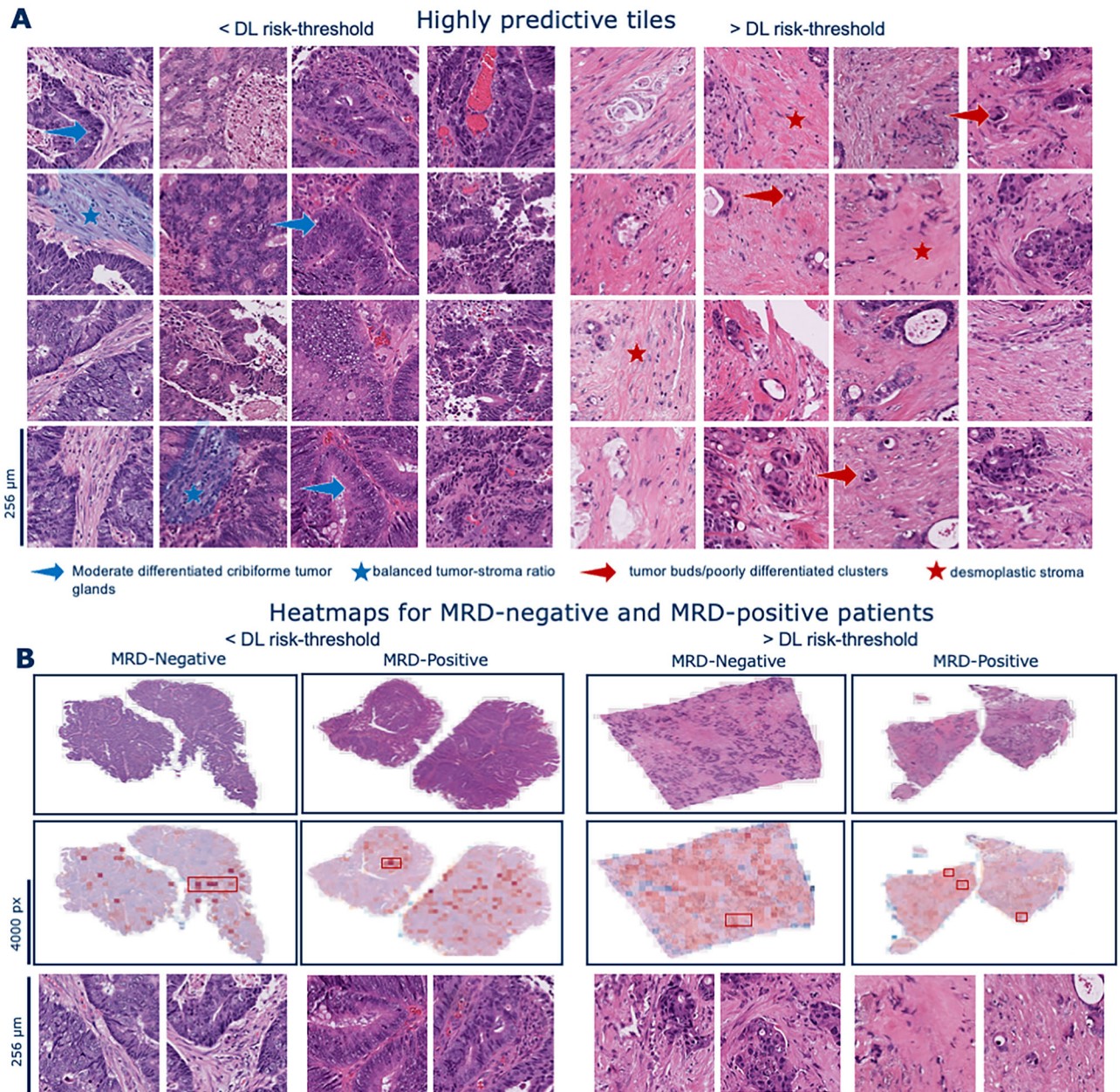

**Fig. 4 | DL can identify morphological features linked to prognosis. A** Highly predictive tiles for patients below the DL risk-threshold and above the DL risk-threshold exemplarily with DL score reported. **B** Whole slide patient heatmaps showing the DL prediction score, red indicating high-risk, and blue indicating low-risk. Visualizations are based on a single analysis run and represent typical patterns observed in the dataset. DL=Deep Learning, MRD=molecular residual disease.

evidence suggesting that a DL risk assessment algorithm may indicate therapy efficacy in a real-world setting in CRC. This combined approach may improve patient selection, suggesting a way to restrict ACT to those patients who are most likely to benefit from it. By integrating DL-based risk stratification alongside established and emerging biomarkers such as histopathological features and ctDNA, a more comprehensive, multidimensional risk assessment could be achieved, potentially refining traditional classifications. Regarding feasibility in clinical practice, our DL model is not intended to replace pathologist evaluation but rather to complement it by providing an additional layer of quantitative risk assessment based on whole-slide imaging data. Pathologists' expertise remains crucial, especially for assessing established histopathological features such as vascular invasion and tumor budding, which remain significant independent risk factors. Furthermore, our DL method uses state-of-the-art models, is fully open source and can be reused and adapted by anyone.

A limitation of our exploratory study is that the results should be interpreted as hypothesis-generating rather than conclusive. Integrating our insights into clinical routine requires further evaluation in additional cohorts, ideally in a prospective manner. Despite this, our study, encompassing thousands of patients across different countries, represents one of the largest studies in this field. Moreover, we utilized a state-of-the-art foundation model for digital pathology analysis, UNI[49]. This is particularly relevant for clinical translatability, as the capabilities of foundation models are rapidly advancing, suggesting that further performance gains are conceivable with improved DL models. Nevertheless, medical device approval in Japan, the US, and the European Union requires a static piece of software that cannot be easily updated. This regulatory limitation introduces the risk of the model becoming outdated by the time of clinical approval. A limitation of this study is that other recent foundation models for survival analysis, which may offer enhanced performance, were not evaluated, and

thus direct comparisons are not available. However, since the conclusions of our model are derived from extensive histopathological data and fundamental tumor biology, we believe its prognostic insights will likely remain clinically relevant despite future model iterations. Furthermore, the influence of neoadjuvant treatment on histopathological slides requires careful consideration. Neoadjuvant therapy can alter tumor morphology and stromal composition, potentially affecting the model's interpretability and performance. Further validation in neoadjuvantly treated cohorts will be necessary to ensure consistent performance across diverse clinical scenarios. Additionally, ACT in the GALAXY cohort was administered based on physicians' decisions and patients' preferences rather than a standardized protocol, which may introduce variability in treatment effects and their influence on outcomes. A further limitation is that we solely controlled for the covariates pT and pN staging, M-stage, MSI-status, age, gender, ACT and MRD status and therefore could not account for all potential prognostic variables associated with survival outcomes, and the relatively short median follow-up duration of 22 months represents an additional limitation. Lastly, we did not perform a quantitative analysis of high-attention regions. Future studies could employ segmentation tools to analyze these regions and derive quantitative metrics, such as glandular morphology or nuclear distribution, to further validate the features identified by the DL model.

Despite these limitations, our data show that the excellent prognostic performance of ctDNA in CRC can be further improved by DL-based end-to-end assessment of routine pathology slides. After prospective validation, this approach provides a plausible and comprehensive strategy for relapse risk assessment with potential therapeutic implications.

## Methods

### Ethics statement

The experiments in this study were carried out according to the Declaration of Helsinki and the International Ethical Guidelines for Biomedical Research Involving Human Subjects by the Council for International Organizations of Medical Sciences (CIOMS). The present study also adheres to the "Transparent reporting of a multivariable prediction model for individual prognosis or diagnosis" (TRIPOD) statement.20. The Ethics Board at the Medical Faculty of Technical University Dresden (BO-EK-444102022) and Institutional Review Board of the National Cancer Center Japan (2023-207) approved of the overall analysis in this study. The patient sample collection in each cohort was separately approved by the respective institutional ethics board.

### Patient data acquisition

In this study, we analyzed primary histological whole slide images (WSIs) of H&E-stained tumor tissue of surgically curable CRC from two large cohorts in Germany and Japan (Fig. 1A, B, Supplementary Fig. 1). For most patients, only one slide was available within the scope of the study. The first cohort was the Darmkrebs: Chancen der Verhütung durch Screening Study (DACHS)[50–52], conducted between 2003 and 2010, which includes 1774 WSI's belonging to 1766 patients and served as the training cohort (Supplementary Fig. 1A and Supplementary Table 1). In the DACHS cohort, 43% of participants were female and 57% male. The second cohort, was the GALAXY cohort, an observational arm from the prospective CIRCULATE-Japan study (UMIN000039205, conducted in 2020 and 2024), which includes 1404 primary WSIs from 1404 patients and served as the independent external validation cohort (Supplementary Fig. 1B). In this cohort, 46% of the participants are female, and 54% are male. Patients with synchronous tumors were excluded in the GALAXY cohort, while in the DACHS cohort, they were classified based on the tumor with the highest stage, which also defined tumor location. The GALAXY trial comprised ctDNA data measuring the MRD status at the four weeks post-surgery. The ctDNA

detection method used in the study was based on a tumor informed assay (Signatera, Natera Inc.). Tumor specific somatic single-nucleotide variants (SNVs) were identified via whole-exome sequencing of formalin-fixed, paraffin-embedded tumor tissue, creating a personalized panel of up to 16 tumor-specific variants. Cell-free DNA (cfDNA) extracted from blood plasma was analyzed using a multiplex PCR-based NGS approach to detect ctDNA. MRD positivity was defined by the detection of at least two out of 16 tumor-specific ctDNA variants detected above a predefined threshold based on Natera's method[18,53]. Out of the 1404 patients included in the trial, 239 were MRD-positive and 1165 patients were MRD-negative at the respective 4 weeks interval[15] (Fig. 1B). Only a small proportion of patients in the DACHS (12%) and GALAXY (10%) cohorts received neoadjuvant treatment. Moreover information about adjuvant chemotherapy (ACT) application was available for all the patients in the GALAXY cohort (Supplementary Table 1). Patient sex was self-reported by study participants.

### Image processing and deep learning techniques

**Data preprocessing.** All WSIs were segmented into image patches, each with a edge length of 256 µm and resized to 224 × 224 pixels, resulting in an effective magnification of 1.14 µm per pixel. Tiles with an average number of Canny edges below a threshold of 2 (indicative of background or blurry regions) were excluded from the dataset. The remaining patches were color normalized using the Macenko method[54], in order to avoid stain-associated bias. The preprocessing of the WSIs was done using our open-source pipeline, HIBRID. This pipeline utilizes UNI, a self-supervised, histology-specific pretrained encoder, to transform each image patch into a 1024-dimensional feature vector (Fig. 1). UNI is a general-purpose self-supervised model, pretrained on over 100 million images from more than 100,000 diagnostic H&E-stained WSIs across 20 major tissue types[49]. Preprocessed WSIs - Features are stored in.h5 files, including the coordinates of the patches. One.h5 file corresponds to one WSIs with the dimension of n-tiles x 1024 (Fig. 1D).

**Deep learning model development.** To train and validate our prediction DL-models we used our open-source HIBRID-pipeline (https://github.com/KatherLab/HIBRID). All the features vectors extracted from the patches of each WSI were aggregated into a bag, which was then processed using a Transformer-based Multiple Instance Learning (MIL) model for a patient-level risk-score prediction (Fig. 1D). This process involves projecting the feature vectors from each patch onto a 512-dimensional feature space through a fully connected layer with ReLU activation. A learnable class token (CLS token) is then appended to the sequence, resulting in a new sequence where the CLS token is the first element, followed by the feature vectors of the patches[55]. This new sequence is fed into the transformer module. The transformer architecture is designed with two layers, each compromising two main components: a self attention mechanism and a feedforward network, both wrapped with residual connections and layer Normalization[30]. The self-attention mechanism models the relationship between all patches in the bag, allowing the CLS token to aggregate global information about the WSI. Following this, the feedforward network refines the feature representation through nonlinear transformation. These operations are repeated across both layers, progressively enriching the CLS token. Finally, the updated CLS token, which now encodes the bag-level representation of the WSI is extracted from the sequence and passed through the multilayer perceptron (MLP) head to generate a patient-level risk score (Fig. 1D).

**Model Training.** During each training epoch, 512 tiles are randomly sampled from the WSI. Throughout the training process, this method ensures that the network comprehensively processes all tiles from the WSI. The DACHS cohort was randomly split at patient-level into training, validation, and test sets in a 4:4:2 ratio. The model was trained

on the training set, and the checkpoint with the highest C-index on the validation set was saved. This saved model was subsequently evaluated on the test set and the external validation cohort. The model was trained on an NVIDIA Quadro RTX 8000 GPU with 48 GB of memory, running on a Fedora 37 operating system. Training and evaluation were conducted using the PyTorch library (version 1.12.1) with the Adam optimizer, an initial learning rate of 1e-4, and a batch size of 64. To minimize the risk of overfitting, several established techniques were employed during the training process, including early stopping, L1 regularization, and L2 regularization. Early stopping was configured with a patience of 15, and the weights for L1 and L2 regularization were both set to 0.001. The network was trained for a total of 50 epochs. We adhered to these parameters for training as they had been validated in a prior study[28].

**Visualization.** To interpret our model's output, we generated WSI heatmaps visualizing the tile-level importance and risk scores across the entire slide. Our transformer model processed the patch-level features from the WSI to generate a slide-level risk score. To assess each tile's contribution to the predicted risk score, we computed gradients of the models' output with respect to tile feature vectors using backpropagation. A single importance score for each tile was calculated, by performing element–wise multiplication between the gradients and the feature values, followed by averaging the resulting values across all features. This importance score, which captures the contribution of each tile to the final slide-level risk score, is referred to as the Grad-CAM-like score. Additionally, tile-level prediction scores were obtained by passing each tile individually through the model and normalizing the scores across the WSI. Finally, the Grad-CAM-like scores were weighted by the normalized tile-level prediction scores to derive a comprehensive importance score for each tile. These scores identified the most influential tiles for both categories (Fig. 4A). Heatmaps were created based on these weighted scores, with red indicating high-risk regions and blue indicating low-risk regions. To maintain interpretability, we merged these heatmaps with the original WSI, providing clear insights into the tumor morphology and the model's predictions (Fig. 4B).

**Experimental Design**
In our study, we trained a transformer-based DL model on the DACHS cohort, utilizing clinical data on disease-free survival (DFS) events and DFS time in months to generate patient-level DL-based risk scores (Fig. 1C). DFS marks the time from primary surgery to disease recurrence or death, whichever event occurs first. The median follow-up time for the DACHS cohort was 121 months and 22.6 months for the GALAXY cohort, respectively. The median of the DL-risk score in the training cohort was 0.9357855 and was subsequently used as a threshold for the binarization of the DL-risk scores. The trained DL-model was then externally validated on the GALAXY cohort. The continuous DL-risk scores from the validation cohort were binarized into DL high-risk and DL low-risk categories using the threshold derived from the DACHS cohort. Firstly, we performed a survival analysis comparing the overall DL-derived risk score stratification with the stratification outcomes of MRD status four weeks post-surgery in the GALAXY cohort. Multivariate analysis was conducted using Cox proportional hazard models, including the covariates: age, gender, pathological T-Stage (pT), metastasis stage (M-Stage), ACT, MSI-status and pathological N-Stage (pN)[28] to further evaluate these associations. Secondly, we combined the DL-risk scores with the four-week post-surgery MRD status to analyze survival differences for DL high-risk vs. DL low-risk within the subgroups of MRD-positive and MRD-negative patients. Thirdly, to further stratify the results, we explored the association of ACT with DFS within the DL high-risk and low-risk subgroups among both MRD-positive and MRD-negative patients. In this GALAXY cohort,

patients enrolled in the interventional VEGA and ALTAIR studies were excluded. Therefore, the patients in this cohort received treatment based on clinical practice, guided by clinicopathological evaluations (Taniguchi et al., 2021). All the survival analysis in this study were performed using Kaplan-Meier estimator and log-rank test. Lastly, we performed a morphological analysis to identify histopathological correlations between the DL high-risk and low-risk subgroups, using classification heatmaps to provide interpretability of the DL model's predictions. We evaluated whether our DL model, trained without manual annotations, identifies morphological features of the tumor and TME synergistic to MRD status. Therefore, we visualized highly predictive regions at multiple magnifications in the GALAXY cohort.

**Reporting summary**
Further information on research design is available in the Nature Portfolio Reporting Summary linked to this article.

## Data availability
The respective study Principal Investigators provided the clinical and image data. Clinical, Sequencing and image data from the DACHS and GALAXY studies are available under restricted access due to ethical and legal constraints. For detailed data sharing policies, please refer to the original publications (https://www.nature.com/articles/s41591-024-03254-6#data-availability; https://ascopubs.org/doi/10.1200/JCO.2011.35.9307?url_ver=Z39.88-2003&rfr_id=ori:rid:crossref.org&rfr_dat=cr_pub%20%200pubmed)[15,18,50]. Requests for access to additional de-identified data can be submitted to the corresponding author and will be assessed by the steering committee within approximately 2–3 weeks. Data will be shared solely for the purpose of scientific validation and cannot be reused for other purposes. Source data are provided with this paper.

## Code availability
The pretrained vision encoder UNI is available at https://github.com/mahmoodlab/uni under a CC-BY-NC-ND 4.0 license and may be used only for non-commercial academic research with proper attribution. Access to the model requires prior registration on Hugging Face and acceptance of the terms of use. Our WSI preprocessing pipeline and deep learning model code are publicly available at https://github.com/KatherLab/HIBRID under MIT license.

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

## Acknowledgements

J.N.K is supported by the German Cancer Aid (DECADE, 70115166), the German Federal Ministry of Education and Research (PEARL, 01KD2104C; CAMINO, 01EO2101; SWAG, 01KD2215A; TRANSFORM LIVER, 031L0312A; TANGERINE, 01KT2302 through ERA-NET Transcan), the German Academic Exchange Service (SECAI, 57616814), the German Federal Joint Committee (TransplantKI, 01VSF21048) the European Union's Horizon Europe and innovation program (ODELIA, 101057091; GENIAL, 101096312), the European Research Council (ERC; NADIR, 101114631) and the National Institute for Health and Care Research (NIHR, NIHR203331) Leeds Biomedical Research Center. The views expressed are those of the author(s) and not necessarily those of the NHS, the NIHR or the Department of Health and Social Care. This work was funded by the European Union. Views and opinions expressed are however those of the author(s) only and do not necessarily reflect those of the European Union. Neither the European Union nor the granting authority can be held responsible for them. CIRCULATE-Japan receives financial support from the Japan Agency for Medical Research and Development (grant 19ck0106447h0002-TY). SF is supported by the German Federal Ministry of Education and Research (SWAG, 01KD2215A), the German Cancer Aid (DECADE, 70115166 and TargHet, 70115995) and the German Research Foundation (504101714). The DACHS study (HB, TY, DW and MH) was supported by the German Research Council (BR 1704/6-1, BR 1704/6-3, BR 1704/6-4, CH 117/1-1, HO 5117/2-1, HO 5117/2-2, HE 5998/2-1, HE 5998/2-2, KL 2354/3-1, KL 2354/3-2, RO 2270/8-1, RO 2270/8-2, BR 1704/17-1 and BR 1704/17-2), the Interdisciplinary Research Program of the National Center for Tumor Diseases (NCT; Germany) and the German Federal Ministry of Education and Research (01KH0404, 01ER0814, 01ER0815, 01ER1505A and 01ER1505B).

## Author contributions

C.M.L.L., H.B., J.N.K., and T.Y.o. conceptualized the study. H.B., T.N., T.M., S.M., D.K., H.T., I.T., T.K., E.O., and Y.N. provided clinical and scanned whole slide image data for the GALAXY cohort. T.Y., D.W., H.B.r., and M.H. provided clinical and scanned whole slide image data for the DACHS cohort. C.M.L.L. curated the source data. S.S., X.J., and M.v.T. implemented the deep learning algorithm. S.S. adapted the code for data analysis and visualization. C.M.L.L. and S.S. planned and conducted the experiments. C.M.L.L. interpreted the data. H.S.M., Z.I.C., A.R.M., and J.N.K. assisted with the interpretation of results. N.G.R. and S.F. did the pathological interpretation of the results. C.M.L.L. wrote the first draft of the manuscript. All authors revised the manuscript draft, contributed to the interpretation of the data and agreed to the submission of this paper. Authors with identical initials are differentiated using the second letter of their last name.

## Funding

## Competing interests

C.M.L.L reports honoraria from AstraZeneca. H.B reports research funding from Ono Pharmaceutical and honoraria from Ono Pharmaceutical, Eli Lilly Japan, and Taiho Pharmaceutical. T.M reports honoraria from Chugai, AstraZeneca, and Miyarisan. S.M reports honoraria from Taiho Pharmaceutical Co., Ltd., Chugai Pharmaceutical Co., Ltd., and Eli Lilly CO, Ltd. D.K reports honoraria from Takeda, Chugai, Lilly, MSD, Ono, Seagen, Guardant Health, Eisai, Taiho, Bristol Myers Squibb, Daiichi-Sankyo, Pfizer, Merckbiopharma, and Sysmex: research funding from Ono, MSD, Novartis, Servier, Janssen, IQVIA, Syneoshealth, CIMIC, and Cimicshiftzero. H.T reports speakers' bureau from MSD K.K, Merck Biopharma, Takeda, Taiho, Lilly Japan, Bristol-Myers Squibb Japan, Chugai Pharmaceutical, Ono Yakuhin, Amgen; research funding from Takeda, Daiichi Sankyo. I.T reports speakers' bureau from Medtronic, Johnson &Johnson, Intuitive, Medicaroid, Eli Lilly and research funding from Medtronic, sysmex. S.F has received honoraria from MSD and BMS. T.K reports nothing to declare. E.O reports speakers' bureau from Chugai Pharmaceutical Co., Ltd., Bristol Meyers, Ono Pharmaceutical Co., Ltd., Eli Lilly, Takeda Pharmaceutical Co., Ltd.; research funding from Guardant Health, Inc.; advisory role from Glaxosmithkline plc. Y.N reports advisory role from Guardant Health Pte Ltd., Natera, Inc., Roche Ltd., Seagen, Inc., Premo Partners, Inc., Daiichi Sankyo Co., Ltd., Takeda Pharmaceutical Co., Ltd., Exact Sciences Corporation, and Gilead Sciences, Inc.; speakers' bureau from Guardant Health Pte Ltd., MSD K.K., Eisai Co., Ltd., Zeria Pharmaceutical Co., Ltd., Miyarisan Pharmaceutical Co., Ltd., Merck Biopharma Co., Ltd., CareNet, Inc., Hisamitsu Pharmaceutical Co., Inc., Taiho Pharmaceutical Co., Ltd., Daiichi Sankyo Co., Ltd., Chugai Pharmaceutical Co., Ltd., and Becton, Dickinson and Company, Guardant Health Japan Corp; research funding from Seagen,Inc., Genomedia Inc., Guardant Health AMEA, Inc., Guardant Health, Inc., Tempus Labs, Inc., Roche Diagnostics K.K., Daiichi Sankyo Co., Ltd., and Chugai Pharmaceutical Co., Ltd. T.Y.o reports honoraria from Taiho, Chugai, Eli Lilly, Merck, Bayer Yakuhin, Ono and MSD, and research funding from Ono, Sanofi, Daiichi Sankyo, Parexel, Pfizer, Taiho, MSD, Amgen, Genomedia, Sysmex, Chugai and Nippon Boehringer Ingelheim. S.S. The remaining authors declare no competing interests. J.N.K declares consulting services for Owkin, France; DoMore Diagnostics, Norway; Panakeia, UK; Scailyte, Switzerland; Mindpeak, Germany; and MultiplexDx, Slovakia. Furthermore, he holds shares in StratifAI GmbH, Germany, has received a research grant by GSK, and has received honoraria by AstraZeneca, Bayer, Eisai, Janssen, MSD, BMS, Roche, Pfizer and Fresenius. All the other authors report nothing to declare.

## Additional information

[1]Else Kroener Fresenius Center for Digital Health, Technical University Dresden, Dresden, Germany. [2]Medical Department 1, University Hospital and Faculty of Medicine Carl Gustav Carus, Technische Universität Dresden, Dresden, Germany. [3]National Center for Tumor Diseases Dresden (NCT/UCC), Dresden, Germany. [4]Department of Data Science, National Cancer Center Hospital East, Kashiwa, Japan. [5]Department of Gastroenterology and Gastro-intestinal Oncology, National Cancer Center Hospital East, Kashiwa, Japan. [6]Translational Research Support Office, National Cancer Center Hospital East, Kashiwa, Japan. [7]Department for Visceral, Thoracic and Vascular Surgery, University Hospital and Faculty of Medicine Carl Gustav Carus, Technische Universität Dresden, Dresden, Germany. [8]Pathology, Faculty of Medicine, University of Augsburg, Augsburg, Germany. [9]Bavarian Cancer Research Center (BZKF), Augsburg, Germany. [10]Department of Clinical Oncology, Aichi Cancer Center Hospital, Nagoya, Japan. [11]Department of Surgery, Surgical Oncology and Science, Sapporo Medical University, Sapporo, Japan. [12]Department of Surgery, NHO Osaka National Hospital, Osaka, Japan. [13]Department of Surgery and Science, Graduate School of Medical Sciences, Kyushu University, Fukuoka, Japan. [14]Division of Clinical Epidemiology and Aging Research, German Cancer Research Center (DKFZ), Heidelberg, Germany. [15]Institute of Pathology, University Medical Center Mainz, Mainz, Germany. [16]Medical Oncology, National Center for Tumor Diseases (NCT), University Hospital Heidelberg, Heidelberg, Germany. ✉e-mail: tyoshino@east.ncc.go.jp; kather.jn@tu-dresden.de

