## [Peer Review File · Nature Communications]

HIBRID: Histology-based Risk-stratification with Deep Learning and ctDNA in Colorectal cancer

Corresponding Author: Professor Jakob Nikolas Kather

Version 0:

Reviewer comments:

Reviewer #1

(Remarks to the Author)

Generic comments:

The authors present a DL model based on vision transformers using histopathological images to predict survival rates of CRC patients. Secondly, they evaluate whether combining their DL model's prediction with ctDNA MRD parameters outperforms current risk-stratification system based on ctDNA MRD parameters only. The topic and methodology used is novel and of interest, but the current version of the manuscript needs further structuring and clarification. The manuscript would also benefit from some proofreading.

Overall, the section of Model Development is unclear and poorly explained. It seems the authors used a foundation model as feature extractor, which is subsequently fine-tuned with a Vision Transformer model to predict overall survival as downstream task. There are many details missing regarding the architecture and optimization of the model. Did the authors employ any hyperparameter tuning? A schematic figure of the model architecture could aid to understand and reproduce the results. In the current form, it is difficult to reproduce the methodology of the paper.

All results sections start with methods explanations. This does not help to understand the whole picture of what the authors are analysing. Please, restructure the manuscript accordingly.

Regarding the results section, the authors kindly point out that their model enhances the prognostic value of the ctDNA, although it seems to overlook other factors that could influence survival outcomes and are typically considered for adjuvant treatment after surgery and during follow-up. It might be wonderful for the authors to consider revisiting the analyses to better illustrate the additional benefits of their model compared to the current prognostic classification.

Specific comments:

Abstract

- HR is defined in supplementary materials, but not in the manuscript (Hazard Ratio)

- ACT is defined in Introduction, but not in Abstract

INTRODUCTION:

- None of references 2 and 3 report data on resectable metastatic disease.

- Add some references supporting this sentence "Current prognostication systems for risk assessment, including imaging techniques, clinicopathological features and molecular data such as BRAF, RAS mutational status and microsatellite instability (MSI), are moderate predictors for recurrence risk". Are the authors referring to localized disease?

- L85: more recent guidelines such as the ESMO 2022 should be referred to instead of the paper by O'Connor and colleagues.

- Add references focus on CRC when talking about the role of ctDNA in detecting MRD or monitoring treatment response.

- L95: Rephrase the sentence. Only pathological characteristics and MSI are fully recommended for treatment decisions and are associated with prognosis. RAS and BRAF remain controversial. Immunoscore is a potential tool that requires further validation. On the other hand, ref. 17 is not a clinical guideline.

M&Ms:

- L116: The group uses the DACHS cohort for training, used previously for prognostication in the paper by Jiang and colleagues (DOI: 10.1016/S2589-7500(23)00208-X). Is there any difference between the data published or obtained between both analysis? If yes, can you explain it? The authors refer to the same reference in the bibliography. Can you specify why there are 1,774 WSI's from 1,766 patients?

- L124: The number of patients included in the paper by Kotani et al. are lower than the number of patients reported in this manuscript. Please update the reference or eliminate it from this sentence.

- L121: Please provide clinical and pathological data of the Galaxy cohort patients in suppl. table 1.

Model Development:

- It is noteworthy that 12% of the patients of the DACHS cohort and 10% of the Galaxy cohort received neoadjuvant treatment. How did you deal with it? Have you considered the pathological changes due to therapy? Can you justify this does not affect the model?
- L144-146: What is Marugoto pipeline and what is it used for? Please explain it briefly.
- L145: UNI is a foundation model as stated in the discussion section. Please, refer to it in the Methods as such. Was it pre-trained by the authors or externally? Also, the authors mention it was pretrained with histology-specific images and WSI. What is the difference between them? Please, clarify. By checking reference 33, this statement is not correct.
- L146: Reference 33 should be placed after the explanation of the UNI model instead of after Figure 1 reference.
- L148: Here, the transformer model concept is introduced. Nonetheless, in the visualization section, the authors use Vision Transformer to describe the implemented model. Please keep consistency among concepts throughout the manuscript to ease the comprehension of the section.
- L148-L150: The authors try to first generally explain the concept of a transformer, but by only describing some elements of it (like "self-attention calculates a query-key product"). Please rephrase either to explain all concepts or simplify it to just describe the architecture of the author's model. It is misleading otherwise.
- L155: Regarding the learnable class token: are they using it for prediction? This should be further explained. Also add reference.
- L157-L159: Each transformer layer consists of 2 normalization blocks, and all contain a MLP or just the last one?
- L166: You should change Circulate by Galaxy cohort.

Visualization:

- L170: Be consistent with Vision Transformer / Transformer concept.
- L170: The authors state they combined tile-level scores (presumably attention scores) with Grad-CAM values to generate weighted scores. This is insufficiently explained, it is not clear how this is done and what they exactly mean.

Experimental design:

This section needs further development. All results sections of the manuscript present results from this section and are not explained elsewhere. Please, restructure coherently.

- L183: Is the threshold set by values retrieved from the DL model?
- L187: Correct Kapan-Meier to Kaplan-Meier
- L188: pathological TNM, clinical risk scores (high and low risk), lymphovascular invasion and adjuvant chemo should be included in the multivariate analysis, since are associated with survival.

Data and code availability

- L199: Missing references to "original publications"

RESULTS:

DL stratifies patients by recurrence risk

The authors present evidence indicating that the DL risk score is effective in distinguishing between high-risk and low-risk groups within the Galaxy Cohort (57% vs 79%, HR 2.3). Nevertheless, this score exhibits diminished efficacy following statistical adjustment. It is significant to note that other substantial factors, including TNM and MSI status, as well as adjuvant chemotherapy, were omitted from the multivariate analysis. This raises the question: what is the genuine incremental value of the DL risk score? Tie et al. elucidated that ctDNA, lymphovascular invasion, and clinical risk scores serve as independent prognostic factors in non-metastatic CRC patients, with a hazard ratio exceeding 1.4. Moreover, the authors of the Galaxy study indicated that seroconversion in ctDNA-positive patients post-chemotherapy correlates with improved survival outcomes. I suggest redoing multivariate analysis considering classical clinicopathological risk factors, adjuvant chemo, and molecular biomarkers statistically significant in the univariate analysis.

Additionally, it is noteworthy that 17% of individuals within the Galaxy cohort exhibit stage IV resectable CRC, presumably comprising those who underwent neoadjuvant chemotherapy. Interestingly, this subgroup is evenly distributed between low and high DL risk categories. Furthermore, stage IV patients inherently possess an elevated risk of recurrence. Considering these observations, I propose a concentrated investigation into localized CRC, relegating the overall cohort findings to supplementary materials. Amend the discussion section accordingly.

The assertion, "Together, these data demonstrate that the DL model can significantly stratify patients according to their risk of recurrence," lacks complete accuracy, as further analyses remain to be conducted.

- L202-L206: Move the explanation to M&Ms section.
- L207: FU is referred previously. You can omit it.
- L211: Results (Fig. 1E and 1F) should be put apart from methods. Put them together with ctDNA KM curves, that should not be reported as supplementary material (now depicted in suppl Fig 2A).
- L217: Table 1 should be moved to suppl. Material. It is not relevant for the main objective of the manuscript.

DL stratifies recurrence risk within MRD subgroups

- L232: As the prognostic value of DL risk score is so small compared to MRD, the subgroup "Either high" does not seem useful as it is based on the poor prognosis associated with the positivity of ctDNA analysis.
- L237: the authors should include within the multivariate analysis the prognostic factors as mentioned above.
- L242: only the combination of DL risk score and ctDNA is significant in ctDNA+ patients, thus the data does not support this statement.

DL-based recurrence risk predicts benefit from adjuvant chemotherapy in MRD-negative patients

In this section the authors aim to demonstrate the impact of the DL risk score in the adjuvant setting. However, they do not include the well established risk factors that influence the treatment decision. It is imperative to re-evaluate the analysis, at a

minimum by stratifying according to TNM classification and clinical risk subgroups, to ascertain the genuine added value of the DL score.

- L248: how many patients received adjuvant chemotherapy? Which regimen? Please, add this information.

DL can identify histopathological features linked to prognosis

- L269-L277: Move to M&Ms section

- L293-L297: Move to Discussions

Is the model able to capture the pathological risk factors, such as vascular invasion or tumor budding?

Discussion:

Since the investigation encompasses patients with metastatic disease, even in instances of resection, the primary focus of the initial paragraph within the discussion section should also address this specific subgroup of patients, who inherently possess a heightened risk of relapse. Nevertheless, I think that the model should only focus on primary tumors from patients without metastatic disease. Furthermore, reference 41 does not represent the most suitable source to encapsulate the role of adjuvant chemotherapy in localized CRC. The authors further emphasize the additional value of DL to ctDNA stratification, albeit they fail to account for other independent pathological risk factors such as vascular invasion. It is imperative to deliberate on the potential significance of DL risk stratification within clinical practice and its enhancement of traditional risk classification alongside emerging molecular classifications (ctDNA, Immunocore, etc.).

How feasible is the translation of DL models into clinical settings? Will the pathologist's evaluation no longer be necessary? Or it would be complementary? Please, discuss this issues in this section.

There is an absence of commentary regarding the influence of neoadjuvant treatment on the evaluation of histopathological slides for the purpose of model development and its validation in independent cohorts.

Why may the method be outdated by the time of clinical approval? By learning from so much data, the authors conclusions might still be valid to be applicable into the clinic in some time.

The relatively short median follow-up duration of 22 months constitutes a limitation of the study, as does the absence of data concerning overall survival.

Figures and Tables.

L550: Check and correct the subgroups.

Suppl Table 1. Please edit the table to make it easier to read.

Supplementary Figure 3: Morphological and molecular features for DL low-risk score. Can you provide a detailed description of this figure? I do not realise what is the intention of it.

(Remarks on code availability)

Reviewer #2

(Remarks to the Author)

The manuscript "HIBRID: Histology and ct-DNA based Risk-stratification with Deep Learning" by Loeffler and Bando et al. describes the use of deep learning on histological images to risk-stratify patients based on disease-free survival. The study also focuses on further sub-stratifying patients after post-operative assessment of minimal residual disease using ctDNA and the effect of adjuvant chemotherapy in high-risk and low-risk patients. The study population is large, with 1766 patients used to optimize the DL model and 1404 patients for testing and correlating to ctDNA. The analyses are well thought out and the conclusions are interesting. The DL tool presented in the paper could be of high interest to the research community, with high potential for clinical implementation. I would also like to commend the authors for making the tools publicly available. However, the manuscript would be greatly improved by some additional analyses and reflections in the discussion section. Additionally, the methods are lacking in detail and clarity. Please find my suggestions for major and minor revisions listed below:

Major revisions

1. Several details are lacking from the methods section. These should either be elaborated in the methods or supplementary material.

1a. The ctDNA detection method is not described for anyone not already familiar with the GALAXY cohort. A full description of the Signatera sequencing is not necessary, but the authors should provide a brief overview of the concept of the method.

1b. When were the patients recruited for the DACHS and GALAXY cohorts, respectively?

1c. How were the patients treated? Which factors decided ACT allocation? How were the patients surveilled for recurrence?

1d. When are patients censored in the DFS calculation? Definition of DFS makes more sense in the 'Experimental Design' section of Methods and not 'Patient Data Acquisition'.

1e. Were the WSIs specifically made for this study, or routine slides made for pathological assessments? Were only slides containing primary tumor assessed? How was a slide selected for each patient? Why was only one (on average) slide assessed per patient? Presumably assessing more slides would add more information.

1f. How were patients with synchronous tumors handled?

2. The inclusion/exclusion criteria are poorly described. In Sup Fig 1, what does the following exclusion criteria entail: "DFS_E unavailable", "Missing/Confusing data", "NATERA exclusion". What does "first analysis" and "final analysis" mean for the GALAXY cohort?
3. Why was M category not included in the multivariable model alongside pT and pN? Patients with metastatic disease should have worse prognosis, which should be accounted for in the model.
4. Were other pathological risk factors evaluated (e.g. perineural invasion, lymphovascular invasion, histological subtype, differentiation)? How would including these factors impact the DL score performance in multivariable analysis?
5. It would be interesting seeing stage-stratified analyses. The prognoses of stage II and stage IV patients are widely different, and the performance of both ctDNA and the DL model could thus vary between stages.
6. It is very interesting that the DL groups can stratify within MRD detection groups. Which features separate DL high-risk and low-risk patients within the MRD groups? This could help inform where histopathology gains more information than ctDNA. The analysis of morphological differences between DL high and DL low is cool. Is there an enrichment of certain morphological features between DL categories within MRD categories?
7. There are several potential problems with the ACT analyses. The major problem is that patients were not randomized to receive or not receive ACT. Thus, the factors deciding ACT allocation should be considered, and the limitation of this analysis more thoroughly discussed.
 - 7a. How was ACT allocated? If ACT is not allocated to fragile patients (old, poor performance) they would naturally have a worse prognosis.
 - 7b. Features which may impact ACT allocation (i.e. stage, age, performance etc) should be accounted for in the statistical analysis.
 - 7c. I see no mentioning of accounting for immortal time bias in the survival analysis. As patients receiving ACT cannot do so after death/recurrence, these patients are essentially immortal until ACT is started. Indeed, a lot of the events in the ACT-arms are within the first few months. Do the authors have an explanation for this?
8. Pathological N category clearly carried additional prognostic power – especially in MRD negative patients. I assume this is because the DL model was not informed on pathological slides containing the lymph nodes. It would be interesting for the authors to discuss this further in the discussion. Would it be better to make a model including the lymph node status as well as the DL and MRD results?
9. The tiles in Figure 3 are quite small. I think it would be better to include fewer examples at a larger size, so the pathological features would be easier to discern. I would also recommend annotating the slides for non-pathologists with the morphological features described in the text (line 280-287).
10. How does the DL model handle a slide with both high-risk and low-risk tiles? Would they cancel out each other? This would be interesting to include in the discussion. Also, it would be interesting to touch on the interpretability of the DL model and what that could add in a diagnostic setting.
11. The authors comment that the DL risk scores were similarly distributed between MSI, BRAF, and RAS statuses, which indicates that the DL model "independently detects and accounts for additional prognostically relevant morphological features". What is the proof of that? If the DL score had no clinical relevance, the result would presumably be the same? Without a clinical endpoint in this analysis, I don't think it can be readily interpreted.

Minor revisions

12. Line 48 (abstract): "Spatial information about the tumor and its microenvironment is not directly measured by ctDNA". I would argue it is not at all measured by ctDNA. I suggest revising, as this sentence is confusing.
13. Line 76-77, the authors write: "Despite advancements in surgical and adjuvant therapies, recurrence rates exceed 30% and 60%^{2,3}, respectively". What does respectively refer to? Neither of the references mentions a 60% recurrence rate. Additionally, ref 2 is based on data from 2015. Ref 3 shows a drop in recurrence rate over the years, with a 5yr cumulative incidence of recurrence of only 25% in recent (2014-2019) stage III patients and 17% for stage II. The introduction (and abstract) should be revised accordingly.
14. Supplementary Table 1 would be a lot easier to read with cell borders.
15. Line 124-125+129-130: These are results and should not be included in the methods section.
16. The DACHS cohort was split into training, validation and test. Are results on the Test dataset ever used/shown in the manuscript?
17. Line 166: should "CIRCULATE data cohort" be "GALAXY data cohort"?
18. Line 199: "For detailed data sharing policies, please refer to the original publications". The authors should cite the publications mentioned here.

19. Line 330: the authors state that their study encompasses patients “across different ethnicities”. I assume this refers to the fact that the DACHS cohort is German and GALAXY is Japanese? As ethnic information is not included in any of the tables/analyses, I suggest the authors remove this statement.

20. Line 296-297: This does not belong in a result section. This is for the discussion.

21. Number of patients in each subgroup in the boxplots of Fig 3 C-E should be noted on the figure or in the legend.

22. Line 337-338: This call to action for regulators and policymakers seems out of place in a scientific paper.

Despite my many comments, I think the manuscript is of high quality and worthy of publication. My feedback is meant to further enhance the manuscript, which I believe will be of wide interest to multiple research niches.

(Remarks on code availability)

I have not reviewed the code, as I am not qualified to assess it.

Reviewer #4

(Remarks to the Author)

(Remarks on code availability)

Reviewer #5

(Remarks to the Author)

This manuscript presents a method utilizing vision transformers to predict disease-free survival (DFS) from histological H&E-stained WSIs of patients with resectable stage II-IV CRC. The proposed approach stratifies CRC patients into high- and low-risk groups and offers an alternative to ctDNA-based recurrence prediction, which can be time-intensive and costly. The method demonstrated efficiency and statistical significance, with training on a large cohort (DACHS) and independent validation on another sizable cohort (GALAXY). Moreover, they proved that combining the deep learning model outputs with MRD status derived from ctDNA further enhances patient stratification.

The work addresses an unmet clinical need, and is interesting on a more technical level as H&E predictors of ctDNA biomarkers have not been studied in detail in the literature. The framework is robust, the results are significant, and the datasets used for the study are large and of interest for the community.

My major comments are regarding the methodology, both for the prediction and the interpretability analysis, both of which seem suboptimal.

Major comments

1. Patient risk stratification (survival analysis), is a well-established task in WSI analysis. The proposed method adopts a MIL framework, using UNI as the tile encoder and a shallow ViT as the aggregator to generate final WSI embeddings for Cox loss computation. This is a non-standard framework chosen by the authors, without clearly justifying that choice. Many well-known MIL-based WSI classification models, such as CLAM and TransMIL, have been shown by other studies to be robust and efficient across various WSI analysis tasks. While these models were originally trained using ImageNet-pretrained tile encoders, their encoders can be easily replaced with any foundation model such as UNI for better performance. Recently published WSI-level foundation models like Gigapath, PRISM and TITAN are capable of survival prediction without requiring additional training of a shallow ViT used in the proposed method. It would be useful to understand why this approach was used, its advantages, and performance with respect to other validated methods.

2. Regarding the interpretation analysis, the study presents only a few cases with cropped patches exhibiting high attention scores, and the main conclusion that “DL can identify histopathological features linked to prognosis” is based solely on these selected patches and simple qualitative analysis. A more robust approach would involve conducting a quantitative analysis of high-attention regions across all cases in different groups. For instance, publicly available gland and nuclei segmentation models could be utilized to segment these regions, and morphological irregularity indicators (shape of glands, distribution of tumor nuclei, etc.) derived from the segmentation results could be defined and used for statistical interpretation analysis.

Minor comments

1. The title “HIBRID: Histology and ct-DNA based Risk-stratification with Deep Learning” suggests a multi-modal deep learning framework (integrating histology and ctDNA) for pan-cancer patient risk stratification. However, the model presented in the manuscript is trained solely on histology images to predict risk scores, with the ctDNA component limited to plotting

Kaplan-Meier (KM) curves for MRD subgroups. Additionally, the study focuses exclusively on recurrence prediction in colorectal cancer (CRC) patients, rather than a pan-cancer risk-stratification approach. I recommend that the authors revise the title to more accurately reflect the scope and subject of the study.

(Remarks on code availability)

Reviewer #6

(Remarks to the Author)

(Remarks on code availability)

Version 1:

Reviewer comments:

Reviewer #1

(Remarks to the Author)

Generic comments:

The authors have appropriately responded to the reviewer's comments by incorporating the necessary revisions in the Methods, especially regarding the UNI feature extraction, the subsequent transformer-based model development and training, and Results sections. The clarifications and adjustments provided contribute to a better understanding of the methodology and interpretation of the findings.

Specific comments:

- Substitute reference 13 by Cervantes et al for Argiles et al., that refers to localised disease.
- L189: Add reference to figure 4 on this section.
- L257: There are discrepancies between the text and the figure 2C. Which are the correct one?
- L336: I suggest including mCRC unless the benefit from adjuvant chemo remain controversial
- Suppl table 1. As you have the data, I suggest including in the table the information about adjuvant chemo in the Galaxy cohort stratified by DL risk.
- Suppl table 2. There is a typo "adjuvant" instead of "adjuvant"
- In the discussion, please emphasize the importance of your model to identify those patients that could be safely managed without ACT.

(Remarks on code availability)

Reviewer #2

(Remarks to the Author)

The authors have improved their manuscript considerably. However, there are still some points, which were not addressed sufficiently in the new version of the manuscript. I have outlined them below with numbers referring to my original comments.

1e+1f: Thank you to the authors for explaining to me the origin of the slides for WSIs and how synchronous tumors were handled. This explanation should also be included in the methods section.

3: The authors refrain from including M category in their multivariable model, grouping it with other prognostic markers such as tumor budding and vascular invasion, and stating that they limit their analysis to "the clinically most relevant factors". However, I would argue that metastatic disease is one of the MOST relevant factors for patient prognosis, as patients with stage IV disease have both higher recurrence rates and reduced survival. M category is arguably more clinically relevant than sex and age, which was included in the model by the authors. Therefore, metastatic disease should not be grouped with tumor budding and other niche histopathological risk factors. Citing another study (Jiang et al. 2024), where these factors were not included, is not a satisfactory argument, especially considering the overlap of authors between the cited study and the current manuscript. Additionally, as the authors have included stage-stratified analyses of the DL algorithm performance, the M category information is available and should be included in prognostic analyses. The authors argue that the DL model incorporates several features into a single output, but that is exactly why it is interesting if the performance of this single output is better than each of the factors individually. In their stage-stratified analysis, the authors show less prognostic power of the DL algorithm in stage IV patients. This further highlights the importance of including M category as a factor in the multivariable model.

7b: I appreciate the inclusion of the supplementary table outlining differences between ACT-treated and non-ACT-treated patients. However, the differences should be used in the discussion to reflect on whether these patients actually benefit from ACT treatment or the worse outcome for non-treated patients is simply because this group includes more fragile patients (higher ECOG-status, older, higher proportion of stage IV patients).

7c: I think the authors may have misunderstood my comment, and I apologize for the lack of clarity. The landmark analysis included to account for immortal time bias should instead have been made for the analyses regarding ACT treatment (Fig 3C-F), and not on the DL algorithm alone. It is when splitting the patients based on receiving ACT, immortal time bias may become an issue, as this is where patient have to stay alive until receiving ACT (typically 2-3 months after surgery). Patients in the ACT group are not at risk for an event until ACT has started and are therefore "immortal".

13: The authors have modified the sentence in the introduction to the following: "Despite advancements in surgical and adjuvant therapies, recurrence rates exceed 30% and 60% in resectable metastatic CRC". I still think the wording is very confusing. It should be specified that the 30% refers to non-metastatic (I assume?) and the 60% to resectable metastatic. However, the authors should also change the "30%" figure, as this is way too high. According to the papers, the authors have cited, the recurrence rates are 12-30%, with 30% only being valid for stage III rectal cancers. Therefore, it does not "exceed 30%".

(Remarks on code availability)

Reviewer #4

(Remarks to the Author)

(Remarks on code availability)

Reviewer #5

(Remarks to the Author)

The authors have addressed my questions in part.

1. The conclusion here is that even though there are now established, powerful foundation models that can be used for survival modelling, a different framework was used in this study, which one could reasonably expect may have resulted in lower performance -- by how much is not known because no experiments were done. This at least requires an explanation in the limitations section.

2. The authors have not addressed this point. Given the subjectivity of this patch selection, it is not possible to state generally that the model is generally "able to highlight regions linked to prognostic features". Clearly some are, but we do not know how common this is, whether it was a coincidence, or a biased choice. The objective of providing some explainability is good, but these results are not systematic enough to support such statements. Qualitative evaluation of the patches is acceptable, but there needs to be a convincing sampling process in order to get to a solid conclusion.

(Remarks on code availability)

Reviewer #6

(Remarks to the Author)

(Remarks on code availability)

I didn't run the code. But the instructions are clear and easy to follow. Should be enough to install and run the application.

Version 2:

Reviewer comments:

Reviewer #2

(Remarks to the Author)

I congratulate the authors on a fine piece of work. I have no further comments, and wish them good luck with their

publication.

(Remarks on code availability)

Reviewer #6

(Remarks to the Author)

The authors have addressed all my concerns. I have no additional comments.

(Remarks on code availability)

I didn't run the code. But the instructions are clear and easy to follow. Should be enough to reproduce.

1 Reviewer #1 (Remarks to the Author)

Generic comments:

The authors present a DL model based on vision transformers using histopathological images to predict survival rates of CRC patients. Secondly, they evaluate whether combining their DL model's prediction with ctDNA MRD parameters outperforms current risk-stratification system based on ctDNA MRD parameters only. The topic and methodology used is novel and of interest, but the current version of the manuscript needs further structuring and clarification. The manuscript would also benefit from some proofreading.

Response: Thank you very much for your valuable feedback and for recognizing the novelty and relevance of our work. In response to your insightful comments, we have carefully revised our manuscript to improve its structure and clarity. Specifically, the Methods section was rearranged and rephrased to provide a clearer explanation of our approach. Moreover we added a stage specific analysis. We hope these changes have strengthened the manuscript.

Overall, the section of Model Development is unclear and poorly explained. It seems the authors used a foundation model as feature extractor, which is subsequently fine-tuned with a Vision Transformer model to predict overall survival as downstream task. There are many details missing regarding the architecture and optimization of the model. Did the authors employ any hyperparameter tuning? A schematic figure of the model architecture could aid to understand and reproduce the results. In the current form, it is difficult to reproduce the methodology of the paper.

Response: We agree with you and therefore completely revised the Methods section to make it more comprehensible and clearer. Additionally, we added a new Figure 1D to illustrate the model architecture. Hyperparameter tuning was not performed, as we used the same hyperparameters explored and optimized in the foundational work by (Jiang et al. 2024).

All results sections start with methods explanations. This does not help to understand the whole picture of what the authors are analysing. Please, restructure the manuscript accordingly.

Response: Thank you for pointing this out. We shortened the introductory parts of each result section focusing more on the findings of the study, while reducing methodological details and avoiding redundancy.

Regarding the results section, the authors kindly point out that their model enhances the prognostic value of the ctDNA, although it seems to overlook other factors that could influence survival outcomes and are typically considered for adjuvant treatment after surgery and during follow-up. It might be wonderful for the authors to consider revisiting the analyses to better illustrate the additional benefits of their model compared to the current prognostic classification.

Response: Thank you for your thoughtful feedback. We acknowledge the importance of considering multiple factors influencing survival outcomes. In our analysis, we controlled for key clinicopathological factors commonly used in prognostic assessments, including pT, pN staging, age, gender and MRD status assessed by ctDNA. However, it would be impossible to account for all potential prognostic variables. We have now addressed this point more explicitly in the revised *Limitations* section from line 389 to 392. We hope this clarification addresses your concerns. Moreover we added a stage-specific analysis and patient characteristic analysis comparing the distributions of those patients receiving ACT and those who did not.

Specific comments:

Abstract

- HR is defined in supplementary materials, but not in the manuscript (Hazard Ratio)
- ACT is defined in Introduction, but not in Abstract

Response: Indeed we updated the abbreviations for Hazard Ratio and ACT. Thank you for finding this error.

INTRODUCTION:

- None of references 2 and 3 report data on resectable metastatic disease.

Response: We added the references (Oki et al. 2024; Kanemitsu et al. 2021) to address this comment.

- Add some references supporting this sentence “Current prognostication systems for risk assessment, including imaging techniques, clinicopathological features and molecular data such as BRAF, RAS mutational status and microsatellite instability (MSI), are moderate predictors for recurrence risk”. Are the authors referring to localized disease?

Response: We have now added the following references to support the statement: (García-Alfonso et al. 2020; Chen et al. 2021; Xiong et al. 2023). Additionally, we clarify that our analysis specifically refers to stage II - resectable metastatic stage IV colorectal cancer.

- L85: more recent guidelines such as the ESMO 2022 should be referred to instead of the paper by O'Connor and colleagues.

Response: Thank you for the suggestion. We have replaced the citation of O'Connor et al. with the more recent papers by (Baxter et al. 2022; Cervantes et al. 2023)

- Add references focus on CRC when talking about the role of ctDNA in detecting MRD or monitoring treatment response.

Response: We agree and have added CRC-specific references (Moding et al. 2021; Nakamura et al. 2024).

- L95: Rephrase the sentence. Only pathological characteristics and MSI are fully recommended for treatment decisions and are associated with prognosis. RAS and BRAF remain controversial. Immunoscore is a potential tool that requires further validation. On the other hand, ref. 17 is not a clinical guideline.

Response: We have revised the section in the introduction (lines 99 to 101) for clarity and adjusted the references to (Luchini et al. 2019; Koncina et al. 2020) to better align with clinical guidelines.

M&Ms:

- L116: The group uses the DACHS cohort for training, used previously for prognostication in the paper by Jiang and colleagues (DOI: 10.1016/S2589-7500(23)00208-X). Is there any difference between the data published or obtained between both analysis? If yes, can you explain it? The authors refer to the same reference in the bibliography. Can you specify why there are 1,774 WSI's from 1,766 patients?

Response: Correct, we used the DACHS as a training, as in the paper by (Jiang et al. 2024) . However, we excluded stage I CRC patients to ensure alignment with the patient population

included in the GALAXY trial, which focused on stage II-IV patients. Other than this exclusion, there is no difference in the data used between both analyses. Additionally, the discrepancy between the number of WSIs (1,774) and patients (1,766) arises because some patients have multiple WSIs included in the analysis.

- L124: The number of patients included in the paper by Kotani et al. are lower than the number of patients reported in this manuscript. Please update the reference or eliminate it from this sentence.

Response: Agreed. While starting the study we referred to the initial primary dataset, however patient number increased as more patients were enrolled. We updated the citation to the latest recently published study from the GALAXY trial from Nakamura et al 2024 in Nature medicine, comprising 2,240 patients, published after our analysis started.

- L121: Please provide clinical and pathological data of the Galaxy cohort patients in suppl. table 1.

Response: We added the data for the GALAXY cohort in the Supplementary Table 1.

Model Development:

- It is noteworthy that 12% of the patients of the DACHS cohort and 10% of the Galaxy cohort received neoadjuvant treatment. How did you deal with it? Have you considered the pathological changes due to therapy? Can you justify this does not affect the model?

Response: Thank you for your question. The proportion of patients receiving neoadjuvant treatment was relatively low (12% in the DACHS cohort and 10% in the GALAXY cohort). While no prior studies have specifically investigated the impact of neoadjuvant therapy on our model, we acknowledge this as a potential limitation and have explicitly addressed it in the *Limitations* section (lines 404–406).

- L144-146: What is Marugoto pipeline and what is it used for? Please explain it briefly.

Response: Since we thoroughly revised the *Model Development* section, the term *Marugoto* no longer appears in the manuscript. We acknowledge that its mention was confusing, as it referred to our internal image processing pipeline. The current version is based on some of its core functions, such as slide loading. We have now replaced it with the updated pipeline, *HIBRID*, which has been made publicly available at: <https://github.com/KatherLab/HIBRID>

- L145: UNI is a foundation model as stated in the discussion section. Please, refer to it in the Methods as such. Was it pre-trained by the authors or externally? Also, the authors mention it was pretrained with histology-specific images and WSI. What is the difference between them? Please, clarify. By checking reference 33, this statement is not correct.

Response: Thank you for your comment. We have updated the Methods section (lines 144-149) to clarify that the UNI model is an externally pretrained foundation model, available on Hugging Face, therefore not pretrained by us. Regarding the distinction between histology-specific images and WSI: UNI was pretrained on patches extracted from over 100,000 diagnostic WSIs (Whole Slide Images). The term "histology-specific images" was used imprecisely in the manuscript and has now been corrected to reflect that the patches come directly from the WSIs.

- L146: Reference 33 should be placed after the explanation of the UNI model instead of after Figure 1 reference.

Response: Thank you. We updated the section and renewed the references accordingly.

- L148: Here, the transformer model concept is introduced. Nonetheless, in the visualization section, the authors use Vision Transformer to describe the implemented model. Please keep consistency among concepts throughout the manuscript to ease the comprehension of the section.
- L148-L150: The authors try to first generally explain the concept of a transformer, but by only describing some elements of it (like “self-attention calculates a query-key product”). Please rephrase either to explain all concepts or simplify it to just describe the architecture of the author’s model. It is misleading otherwise.
- L155: Regarding the learnable class token: are they using it for prediction? This should be further explained. Also add reference.
- L157-L159: Each transformer layer consists of 2 normalization blocks, and all contain a MLP or just the last one?

Response: Thank you for the comments referring to the transformer architecture, which we have revised in the method section of our manuscript. We agreed and therefore decided to focus on the architecture of our transformer model used in the study. The CLS token is used for the Risk score Prediction see publication from (Dosovitskiy et al. 2020). We also added a new Figure 1D hopefully making it clearer.

- L166: You should change Circulate by Galaxy cohort.

Response: Indeed this could be misleading, therefore we changed it.

Visualization:

- L170: Be consistent with Vision Transformer / Transformer concept.
- L170: The authors state they combined tile-level scores (presumably attention scores) with Grad-CAM values to generate weighted scores. This is insufficiently explained, it is not clear how this is done and what they exactly mean.

Response: We apologize for the misunderstanding. The weighted Grad-CAM-like score combines:

1. **A Grad-CAM-like score, which represents feature-level contributions to the overall slide-level prediction.**
2. **A tile-level score, which reflects the model’s independent risk prediction for each tile.**

These two scores are multiplied to produce the weighted Grad-CAM-like score, identifying the most influential tiles with the greatest impact on the final risk score. This process is integral to generating the heatmaps and selecting top tiles. We have clarified this in the revised “Visualisation” section for better understanding.

Experimental design:

This section needs further development. All results sections of the manuscript present results from this section and are not explained elsewhere. Please, restructure coherently.

- L183: Is the threshold set by values retrieved from the DL model?
- L187: Correct Kapan-Meier to Kaplan-Meier
- L188: pathological TNM, clinical risk scores (high and low risk), lymphovascular invasion and adjuvant chemo should be included in the multivariate analysis, since are associated with survival.

Response: Thank you for your valuable comments. We have removed the introductory methodological sentences from the results section and integrated them into the "Experimental

Design" section. Additionally, we have aligned the description of the methodological steps with the structure of the results section. This has made the layout much clearer and more consistent with the presented results. We hope this improves readability for the audience.

In addition, we have addressed the specific points mentioned:

- **L183: Yes, the threshold is based on values retrieved from the DL model.**
- **L187: The typo has been corrected, changing "Kapan-Meier" to "Kaplan-Meier."**
- **L188: As discussed above, we included key factors such as pT, pN staging, age, gender, and MRD status (ctDNA) in our analysis. While we acknowledge the relevance of additional variables, it is not feasible to account for all, as noted in the revised *Limitations* section. Moreover data for clinical risks core and lymphovascular invasion was not available for us.**

Data and code availability

- L199: Missing references to "original publications"

Response: We added the latest publication from the GALAXY cohort trial from Nakamura et al in 2024.

RESULTS:

DL stratifies patients by recurrence risk

The authors present evidence indicating that the DL risk score is effective in distinguishing between high-risk and low-risk groups within the Galaxy Cohort (57% vs 79%, HR 2.3). Nevertheless, this score exhibits diminished efficacy following statistical adjustment. It is significant to note that other substantial factors, including TNM and MSI status, as well as adjuvant chemotherapy, were omitted from the multivariate analysis. This raises the question: what is the genuine incremental value of the DL risk score? Tie et al. elucidated that ctDNA, lymphovascular invasion, and clinical risk scores serve as independent prognostic factors in non-metastatic CRC patients, with a hazard ratio exceeding 1.4. Moreover, the authors of the Galaxy study indicated that seroconversion in ctDNA-positive patients post-chemotherapy correlates with improved survival outcomes. I suggest redoing multivariate analysis considering classical clinicopathological risk factors, adjuvant chemo, and molecular biomarkers statistically significant in the univariate analysis.

Response: Thank you for your detailed comment. As you noted, there is an extensive list of prognostic factors, including budding, vascular invasion, tumor differentiation, immunoscore, MSI, BRAF, KRAS, and many others. In our study, we made the deliberate choice to limit the multivariate analysis to the clinically most relevant factors, as has been done in previous studies, such as Jiang et al. (Jiang et al. 2024) One of the strengths of our deep learning model is its ability to automatically learn and integrate all relevant tissue parameters into a single output. While our DL model is not performing magic, it likely incorporates many of these morphological factors by identifying and combining them in a way that reflects their collective contribution to the outcome. The key advantage is that the model does this integration without us explicitly defining what to look for, streamlining the analysis and allowing for an unbiased synthesis of complex tissue characteristics.

Additionally, it is noteworthy that 17% of individuals within the Galaxy cohort exhibit stage IV resectable CRC, presumably comprising those who underwent neoadjuvant chemotherapy. Interestingly, this subgroup is evenly distributed between low and high DL risk categories.

Furthermore, stage IV patients inherently possess an elevated risk of recurrence. Considering these observations, I propose a concentrated investigation into localized CRC, relegating the overall cohort findings to supplementary materials. Amend the discussion section accordingly.

The assertion, "Together, these data demonstrate that the DL model can significantly stratify patients according to their risk of recurrence," lacks complete accuracy, as further analyses remain to be conducted.

Response: Thank you for the thoughtful comment. We acknowledge that stage IV resectable CRC inherently carries an elevated risk of recurrence, and we note that the distribution of stage IV patients who received neoadjuvant chemotherapy (n=117) versus those who did not (n=123) is nearly equal in the GALAXY cohort. However, excluding stage IV resectable patients from the primary analysis would not align with our study's aim, which focuses on surgically curable CRC (stage II-IV). By including stage IV resectable patients, we capture a clinically relevant subset often managed with curative intent, thereby providing a more comprehensive understanding of the DL model's prognostic capabilities. Nevertheless, we appreciate the reviewer's perspective and therefore include a stage-specific analysis in the Result section and supplementary materials (Supplementary Figure 3), as suggested.

- L202-L206: Move the explanation to M&Ms section.

Response: Done

- L207: FU is referred previously. You can omit it.

Response: Done.

- L211: Results (Fig. 1E and 1F) should be put apart from methods. Put them together with ctDNA KM curves, that should not be reported as supplementary material (now depicted in suppl Fig 2A).

Response: Thank you for this idea. We updated Figure 1 adding a more detailed method part of the Transformer model and created a new Figure 2 visualizing the Kaplan Meier plots for the DL-risk score and ctDNA stratification by DFS and Cox multivariate analysis.

- L217: Table 1 should be moved to suppl. Material. It is not relevant for the main objective of the manuscript.

Response: Good Point. We merged Table 1 into the new Supplementary Table 1 containing the patient characteristics for all the patients in the DACHS and GALAXY cohort and now also stratified into DL high-risk and DL low-risk.

DL stratifies recurrence risk within MRD subgroups

- L232: As the prognostic value of DL risk score is so small compared to MRD, the subgroup "Either high" does not seem useful as it is based on the poor prognosis associated with the positivity of ctDNA analysis.

Response: We acknowledge the reviewer's concern; however, we have decided to retain the "Either high" subgroup to ensure completeness of the analysis. It allows for a comprehensive assessment of the combined prognostic value of both the DL risk score and ctDNA positivity, despite the stronger prognostic impact of ctDNA alone.

- L237: the authors should include within the multivariate analysis the prognostic factors as mentioned above.

Response: See above.

- L242: only the combination of DL risk score and ctDNA is significant in ctDNA+ patients, thus the data does not support this statement.

Response: Indeed. We have clarified the statement in the manuscript for greater accuracy. While the DL risk score was not a significant independent prognostic factor in the ctDNA-negative group in the multivariate analysis, it was significant in the univariate analysis. To better reflect this, we have rephrased the conclusion see line 286-287

DL-based recurrence risk predicts benefit from adjuvant chemotherapy in MRD-negative patients. In this section the authors aim to demonstrate the impact of the DL risk score in the adjuvant setting. However, they do not include the well established risk factors that influence the treatment decision. It is imperative to re-evaluate the analysis, at a minimum by stratifying according to TNM classification and clinical risk subgroups, to ascertain the genuine added value of the DL score.

Response: To address this thoroughly, we have now included a stage-specific analysis to better account for established risk factors influencing treatment decisions. The new findings have been incorporated into the Results section (line 305-307) and Supplementary Figure (3). We believe this addition clarifies the added value of the DL risk score in the adjuvant setting.

- L248: how many patients received adjuvant chemotherapy? Which regimen? Please, add this information.

Response: Thank you for the comment. In this dataset, adjuvant chemotherapy (ACT) was administered based on clinicopathological risk factors, including pathological TNM staging. Moreover physicians' decisions and patients' preferences, reflecting real-world clinical practice guided by clinicopathological evaluations influenced the treatment choice (Taniguchi et al., 2021). We have clarified this in the Methods section (line 224-228) and added the quantitative information about adjuvant chemotherapy allocation to Supplementary Table 1. Moreover we added a note in the *Limitations* section to acknowledge the potential impact of this variability on our analysis.

DL can identify histopathological features linked to prognosis

- L269-L277: Move to M&Ms section

Response: Done.

- L293-L297: Move to Discussions

Response: Done. We incorporated the last part of the result section into the discussion section from line 352 to 358

Is the model able to capture the pathological risk factors, such as vascular invasion or tumor budding?

Response: Very good question. The model was not explicitly trained on specific pathological features, such as vascular invasion or tumor budding. However, deep learning (DL) models are

designed to automatically learn and integrate a wide range of morphological factors—both known and potentially unknown—into their predictions. This likely includes features such as vascular invasion, tumor budding, tumor differentiation, immunoscore, and stromal patterns, without requiring explicit predefinition of these factors.

Discussion:

Since the investigation encompasses patients with metastatic disease, even in instances of resection, the primary focus of the initial paragraph within the discussion section should also address this specific subgroup of patients, who inherently possess a heightened risk of relapse.

Response: We thank the reviewer for this insightful comment. We have revised the opening paragraph of the Discussion section and added the sentence: notably in cases of metastatic disease (line 335) to more clearly address the subgroup of patients. We hope this modification ensures a more accurate reflection of the patient population.

Nevertheless, I think that the model should only focus on primary tumors from patients without metastatic disease.

Response: As discussed previously above. We believe that stratifying patients by recurrence risk in a cohort that reflects the full spectrum of surgically curable CRC—including stage IV resectable disease—better aligns with real-world clinical scenarios and highlights the model's generalizability.

Furthermore, reference 41 does not represent the most suitable source to encapsulate the role of adjuvant chemotherapy in localized CRC.

Response: We have updated the references to better reflect the role of adjuvant chemotherapy in localized CRC. The revised citations now include (Chau and Cunningham 2006), (Taieb and Gallois 2020; Rebutz et al. 2020), and the review article by (Chan and Chee 2019)

The authors further emphasize the additional value of DL to ctDNA stratification, albeit they fail to account for other independent pathological risk factors such as vascular invasion. It is imperative to deliberate on the potential significance of DL risk stratification within clinical practice and its enhancement of traditional risk classification alongside emerging molecular classifications (ctDNA, Immunocore, etc.).

How feasible is the translation of DL models into clinical settings? Will the pathologist's evaluation no longer be necessary? Or it would be complementary? Please, discuss this issues in this section.

There is an absence of commentary regarding the influence of neoadjuvant treatment on the evaluation of histopathological slides for the purpose of model development and its validation in independent cohorts.

Why may the method be outdated by the time of clinical approval? By learning from so much data, the authors conclusions might still be valid to be applicable into the clinic in some time.

The relatively short median follow-up duration of 22 months constitutes a limitation of the study, as does the absence of data concerning overall survival.

Response: We thank the reviewer for these valuable insights. In response, we have expanded the discussion to address the following points:

- 1. Pathological Risk Factors: We have added commentary on other independent pathological risk factors such as vascular invasion and their potential relevance to risk stratification. (see line 378 and 409-411)**

2. **Clinical Integration and Complementarity:** We have elaborated on the feasibility of translating DL models into clinical practice, emphasizing that our model is intended to complement rather than replace pathologist evaluation by providing an additional layer of quantitative analysis (see line 381-385).
3. **Neoadjuvant Treatment Impact:** We now discuss the potential impact of neoadjuvant treatment on histopathological slide analysis and how it may affect both model training and validation in independent cohorts in the limitation section from line 401 to 406.
4. **Model Aging and Clinical Approval:** We clarified why the static nature of current regulatory pathways could limit the adaptability of DL-based models but highlighted how foundational data-driven insights may still retain clinical relevance over time. (see line 378-385 / 398-401)
5. **Follow-up and Survival Data:** We acknowledge the limitation posed by the relatively short median follow-up duration in line 411.

Figures and Tables.

L550: Check and correct the subgroups.

Response: Thank you for this very good observation. We corrected the mistakes.

Suppl Table 1. Please edit the table to make it easier to read.

Response: We increased the font size, highlighted and formatted the table for more readability.

Supplementary Figure 3: Morphological and molecular features for DL low-risk score. Can you provide a detailed description of this figure? I do not realise what is the intention of it.

Response: We thank the reviewer for this observation and decided to exclude Supplementary Figure 3 from the manuscript, as the figure's content did not add substantial value to the conclusions of the study and could be misinterpreted without additional context.

Reviewer #2 (Remarks to the Author)

The manuscript "HIBRID: Histology and ct-DNA based Risk-stratification with Deep Learning" by Loeffler and Bando et al. describes the use of deep learning on histological images to risk-stratify patients based on disease-free survival. The study also focuses on further sub-stratifying patients after post-operative assessment of minimal residual disease using ctDNA and the effect of adjuvant chemotherapy in high-risk and low-risk patients. The study population is large, with 1766 patients used to optimize the DL model and 1404 patients for testing and correlating to ctDNA. The analyses are well thought out and the conclusions are interesting. The DL tool presented in the paper could be of high interest to the research community, with high potential for clinical implementation. I would also like to commend the authors for making the tools publicly available. However, the manuscript would be greatly improved by some additional analyses and reflections in the discussion section. Additionally, the methods are lacking in detail and clarity. Please find my suggestions for major and minor revisions listed below:

Major revisions

1. Several details are lacking from the methods section. These should either be elaborated in the methods or supplementary material.

Response: We thank the reviewer for this comment and indeed we agree with the observation and have thoroughly revised the Methods section. Key methodological details have been expanded, and we have added a new Figure 1D to visually illustrate the model architecture for improved understanding. We hope these revisions sufficiently address your concerns.

1a. The ctDNA detection method is not described for anyone not already familiar with the GALAXY cohort. A full description of the Signatera sequencing is not necessary, but the authors should provide a brief overview of the concept of the method.

Response: We have added a description highlighting the two-step approach of Natera's Signatera assay in the Methods section Patient Data Acquisition (lines 125 to 13) to ensure clarity for readers unfamiliar with the GALAXY cohort.

1b. When were the patients recruited for the DACHS and GALAXY cohorts, respectively?

Response: The GALAXY trial recruited from 5 June 2020 and 30 April 2021 and the DACHS study recruited from 2003-2010. This information can now be found in the "Patient Data Acquisition" Section.

1c. How were the patients treated? Which factors decided ACT allocation? How were the patients surveilled for recurrence?

Response: We thank the reviewer for this important question. In this dataset, adjuvant chemotherapy (ACT) was administered based on physicians' decisions and patients' preferences, reflecting real-world clinical practice guided by clinicopathological evaluations (Taniguchi et al., 2021). We have clarified this in the Methods section (line 224-228) and added the quantitative information about adjuvant chemotherapy allocation to Supplementary Table 1. Moreover we added a note in the *Limitations* section (line 405-408) to acknowledge the potential impact of this variability on our analysis. Regarding surveillance for recurrence, blood samples were collected before surgery and at 4, 12, 24, 36, 48, 72, and 96 weeks postoperatively for ctDNA analysis. Additionally, computed tomography (CT) imaging was performed every six months for up to seven years to monitor for recurrence.

1d. When are patients censored in the DFS calculation? Definition of DFS makes more sense in the 'Experimental Design' section of Methods and not 'Patient Data Acquisition'.

Response: Thank you for pointing this out. We used the lifelines package in Python for performing the Kaplan Meier Analysis. Patients were censored at the time of their last known follow-up if no recurrence or death had occurred. Also we agreed and therefore moved the section to Experimental Design.

1e. Were the WSIs specifically made for this study, or routine slides made for pathological assessments? Were only slides containing primary tumor assessed? How was a slide selected for each patient? Why was only one (on average) slide assessed per patient? Presumably assessing more slides would add more information.

Response: Very good question. The whole-slide images (WSIs) used in this study were routine histopathological slides prepared for standard pathological assessments, not specifically created for this study. Only slides containing primary tumor tissue were included in the analysis. For most patients, only one slide was available within the scope of the study. Consequently, a single representative slide was analyzed for each patient. While we agree that

assessing additional slides could provide more information, the availability of multiple slides per patient was limited in this dataset.

1f. How were patients with synchronous tumors handled?

Response: Patients with synchronous tumors were excluded as part of the predefined exclusion criteria in both the GALAXY trial (Trial Number: UMIN000039205). For the DACHS cohort, patients with synchronous tumors were classified according to the tumor with the highest stage, which also determined the recorded tumor location. However, such cases were rare and unlikely to have influenced the overall results of our study.

2. The inclusion/exclusion criteria are poorly described. In Sup Fig 1, what does the following exclusion criteria entail: “DFS_E unavailable”, “Missing/Confusing data”, “NATERA exclusion”. What does “first analysis” and “final analysis” mean for the GALAXY cohort?

Response: We appreciate the reviewer’s feedback on the clarity of the exclusion criteria. To address this, we have updated the flowchart (Supplementary Figure 1) to ensure a consistent structure between both cohorts. The revised figure now clearly outlines the patient screening process followed by the main exclusion criteria, which were primarily related to early-stage disease, missing data points, or the unavailability of whole-slide images.

3. Why was M category not included in the multivariable model alongside pT and pN? Patients with metastatic disease should have worse prognosis, which should be accounted for in the model.

Response: Thank you for your comment. We acknowledge that there are numerous prognostic factors, including M category, tumor budding, vascular invasion, tumor differentiation and many others. In our study, we made a deliberate decision to limit the multivariable analysis to the clinically most relevant factors, such as pT and pN staging, age, gender, as has been done in similar studies, including Jiang et al. (2024). Our deep learning model complements this by automatically learning and integrating a wide range of morphological and prognostic features into a single output. Although the model does not explicitly account for specific factors, it likely incorporates their collective impact on outcomes, providing an unbiased synthesis of complex tissue characteristics.

4. Were other pathological risk factors evaluated (e.g. perineural invasion, lymphovascular invasion, histological subtype, differentiation)? How would including these factors impact the DL score performance in multivariable analysis?

Response: We recognize the value of incorporating additional pathological risk factors. In our analysis, we controlled for key clinicopathological variables. Factors such as perineural invasion, lymphovascular invasion, histological subtype, and differentiation were not included, as they were unavailable for analysis. While accounting for every potential prognostic variable would be ideal, it is practically challenging and as mentioned previously, we believe this is somehow performed within the DL algorithm. We have explicitly addressed this limitation in the manuscript (“Limitations” section, lines 408-410). Including these additional factors could influence the DL model’s performance, but further studies with larger datasets would be needed to systematically evaluate their impact.

5. It would be interesting seeing stage-stratified analyses. The prognoses of stage II and stage IV patients are widely different, and the performance of both ctDNA and the DL model could thus vary between stages.

Response: We agree and have performed a stage-stratified analysis, which is now included in the new Supplementary Figure 3. The findings have also been added to the Results section (252-255). Notably, in stage II patients, the DL model was able to significantly stratify patients with an HR of 1.87 (CI95% 1.35-2.61; p<0.0005). Thank you for pointing this out!

6. It is very interesting that the DL groups can stratify within MRD detection groups. Which features separate DL high-risk and low-risk patients within the MRD groups? This could help inform where histopathology gains more information than ctDNA. The analysis of morphological differences between DL high and DL low is cool. Is there an enrichment of certain morphological features between DL categories within MRD categories?

Response: We thank the reviewer for this constructive comment. We did not perform a systematic review of histopathological features within MRD categories as this was out of scope for the current study. We acknowledge this Limitation in line 412-415. By reviewing our representative cases (Figure 4 Panel B), we could not identify any relevant differences between MRD-negative / MRD-positive cases < risk threshold. However, we could indeed see that the MRD-positive case > risk threshold did show a poorer differentiation with more single tumor cells/tumor buds compared to the MRD-negative cases < risk-threshold. This high-risk morphology could be linked to MRD-positivity; still further investigations into this direction should be undertaken

7. There are several potential problems with the ACT analyses. The major problem is that patients were not randomized to receive or not receive ACT. Thus, the factors deciding ACT allocation should be considered, and the limitation of this analysis more thoroughly discussed.

7a. How was ACT allocated? If ACT is not allocated to fragile patients (old, poor performance) they would naturally have a worse prognosis.

Response: As the GALAXY study is an observational registry study, the administration of ACT was determined based on clinicopathological risk factors, including pathological TNM staging. Additionally, physicians' decisions and patients' preferences influenced the treatment choices. These factors represent potential limitations of this analysis.

7b. Features which may impact ACT allocation (i.e. stage, age, performance etc) should be accounted for in the statistical analysis.

Response: To address this, we analyzed the patient characteristics between those who received ACT and those who did not. Significant differences were observed in age, ECOG PS, stage, pT, pN, and MSI status. These results have been included in the Results section (307-309) and detailed in Supplementary Table 2.

7c. I see no mentioning of accounting for immortal time bias in the survival analysis. As patients receiving ACT cannot do so after death/recurrence, these patients are essentially immortal until ACT is started. Indeed, a lot of the events in the ACT- arms are within the first few months. Do the authors have an explanation for this?

Response: To address the potential for immortal time bias, we conducted a landmark analysis excluding patients who experienced recurrence within 3 and 6 months (see Supplementary Figure 2A-B), yielding HR of 2.37 and 2.14 respectively. These analyses confirmed consistent trends with the primary analysis, supporting the robustness of our findings. We have briefly mentioned this in the main Results section in line 250-252.

8. Pathological N category clearly carried additional prognostic power – especially in MRD negative patients. I assume this is because the DL model was not informed on pathological slides containing the lymph nodes. It would be interesting for the authors to discuss this further in the discussion. Would it be better to make a model including the lymph node status as well as the DL and MRD results?

Response: Previous studies have shown that the N status can to some extent be predicted from primary histology in colorectal (Kiehl et al. 2021) or in gastric cancer (Hannah S. Muti et al. 2023). Hence, the primary tumor histology inherently contains features which are not exclusively, but partially, predictive of the N status, and it is questionable whether a combination of this information implies a redundancy. Combining a CRC survival prediction model classification with the N status into clinically applicable risk groups, (Jiang et al. 2024) found that this further improved the prognostic value of their model. Further research is needed to determine whether a post-hoc combination of the model output and the N status, as proposed by Jiang et al., or a multimodal training strategy which combines the N status with the WSI for model training result in superior prognostic performance as proposed by Zhou et al (Zhou et al. 2023). We thank the reviewer for the interesting comment and we have added this to the discussion section in lines 354-360.

9. The tiles in Figure 3 are quite small. I think it would be better to include fewer examples at a larger size, so the pathological features would be easier to discern. I would also recommend annotating the slides for non-pathologists with the morphological features described in the text (line 280-287).

Response: We have increased the tile size to make the pathological features more visible. Additionally, we excluded some heatmaps and top tiles to allow for larger and clearer examples. We also annotated the regions and added labels to highlight the morphological features described in the text. Please see the updated Figure 4. We hope these adjustments address your concerns.

10. How does the DL model handle a slide with both high-risk and low-risk tiles? Would they cancel out each other? This would be interesting to include in the discussion. Also, it would be interesting to touch on the interpretability of the DL model and what that could add in a diagnostic setting.

Response: Thank you for your comment. The DL model processes all tiles from a patient slide as a whole, with individual tile scores serving only as local approximations. The model combines these scores nonlinearly to generate a patient-level prediction, ensuring that high- and low-risk tiles do not simply cancel each other out (El Nahhas et al. 2025). References to tile-level scores are intended to aid interpretability but do not fully explain the model's decision-making process at the patient level. We further discussed this important point in line 367-374.

11. The authors comment that the DL risk scores were similarly distributed between MSI, BRAF, and RAS statuses, which indicates that the DL model “independently detects and accounts for additional prognostically relevant morphological features”. What is the proof of that? If the DL score had no clinical relevance, the result would presumably be the same? Without a clinical endpoint in this analysis, I don't think it can be readily interpreted.

Response: The observation that DL risk scores were similarly distributed across MSI, BRAF, and RAS subgroups highlights the model's capacity to extract prognostically relevant morphological features that are not directly tied to these specific molecular statuses. One of the strengths of the DL model is its ability to learn and integrate a wide range of morphological features into a single output without requiring predefined inputs. This approach likely allows the model to capture a combination of known and potentially unknown prognostic factors that

collectively influence outcomes. However, we acknowledge that this analysis does not directly establish the clinical relevance of the DL score in relation to MSI, BRAF, or RAS status. The true value of the DL score is demonstrated through its performance in patient-level survival prediction, where it provides meaningful stratification of recurrence risk, as shown in our main survival analyses.

Minor revisions

12. Line 48 (abstract): "Spatial information about the tumor and its microenvironment is not directly measured by ctDNA". I would argue it is not at all measured by ctDNA. I suggest revising, as this sentence is confusing.

Response: Thank you. To clarify this we rephrased the sentence to: "However, spatial information about the tumor and its microenvironment is not captured by ctDNA. "

13. Line 76-77, the authors write: "Despite advancements in surgical and adjuvant therapies, recurrence rates exceed 30% and 60%^{2,3}, respectively". What does respectively refer to? Neither of the references mentions a 60% recurrence rate. Additionally, ref 2 is based on data from 2015. Ref 3 shows a drop in recurrence rate over the years, with a 5yr cumulative incidence of recurrence of only 25% in recent (2014-2019) stage III patients and 17% for stage II. The introduction (and abstract) should be revised accordingly.

Response: Respectively referred to CRC with resectable metastasis. However we changed the sentence and added new citations to make it clearer.

14. Supplementary Table 1 would be a lot easier to read with cell borders.

Response: We agree, and therefore added the cell borders in Supplementary Table 1.

15. Line 124-125+129-130: These are results and should not be included in the methods section.

Response: We understand the confusion however this data was already available before performing the results, since it was the MRD status measured by ctDNA in the GALAXY cohort.

16. The DACHS cohort was split into training, validation and test. Are results on the Test dataset ever used/shown in the manuscript?

Response: They were used to define the threshold of the DL risk Score, however since the lack of ctDNA availability the data has not been used in the manuscript.

17. Line 166: should "CIRCULATE data cohort" be "GALAXY data cohort"?

Response: Agreed.

18. Line 199:" For detailed data sharing policies, please refer to the original publications". The authors should cite the publications mentioned here.

Response: Done

19. Line 330: the authors state that their study encompasses patients “across different ethnicities”. I assume this refers to the fact that the DACHS cohort is German and GALAXY is Japanese? As ethnic information is not included in any of the tables/analyses, I suggest the authors remove this statement.

Response: We changed the word ethnicities to countries

20. Line 296-297: This does not belong in a result section. This is for the discussion.

Response: Done

21. Number of patients in each subgroup in the boxplots of Fig 3 C-E should be noted on the figure or in the legend.

Response: You are right. We have now added this information to the Plot.

22. Line 337-338: This call to action for regulators and policymakers seems out of place in a scientific paper.

Response: We removed the sentence

Despite my many comments, I think the manuscript is of high quality and worthy of publication. My feedback is meant to further enhance the manuscript, which I believe will be of wide interest to multiple research niches.

Reviewer #4 (Remarks to the Author): Early Career Researcher co-reviewer

Reviewer #5 (Remarks to the Author): Expert in AI, deep learning, and cancer digital pathology

This manuscript presents a method utilizing vision transformers to predict disease-free survival (DFS) from histological H&E-stained WSIs of patients with resectable stage II-IV CRC. The proposed approach stratifies CRC patients into high- and low-risk groups and offers an alternative to ctDNA-based recurrence prediction, which can be time-intensive and costly. The method demonstrated efficiency and statistical significance, with training on a large cohort (DACHS) and independent validation on another sizable cohort (GALAXY). Moreover, they proved that combining the deep learning model outputs with MRD status derived from ctDNA further enhances patient stratification.

The work addresses an unmet clinical need, and is interesting on a more technical level as H&E predictors of ctDNA biomarkers have not been studied in detail in the literature. The framework is robust, the results are significant, and the datasets used for the study are large and of interest for the community.

Response: Thank you.

My major comments are regarding the methodology, both for the prediction and the interpretability analysis, both of which seem suboptimal.

Major comments

1. Patient risk stratification (survival analysis), is a well-established task in WSI analysis. The proposed method adopts a MIL framework, using UNI as the tile encoder and a shallow ViT as the aggregator to generate final WSI embeddings for Cox loss computation. This is a non-standard framework chosen by the authors, without clearly justifying that choice. Many well-known MIL-based WSI classification models, such as CLAM and TransMIL, have been shown by other studies to be robust and efficient across various WSI analysis tasks. While these models were originally trained using ImageNet-pretrained tile encoders, their encoders can be easily replaced with any foundation model such as UNI for better performance. Recently published WSI-level foundation models like Gigapath, PRISM and TITAN are capable of survival prediction without requiring additional training of a shallow ViT used in the proposed method. It would be useful to understand why this approach was used, its advantages, and performance with respect to other validated methods.

Response: We thank the reviewer for this insightful comment. Our shallow ViT model is closely based on the TransMIL architecture (Shao et al. 2021), which has demonstrated robust performance in WSI prediction tasks across multiple studies ((Hannah Sophie Muti et al. 2021; Loeffler et al. 2024; Kather et al. 2020). The decision to use this architecture was based on its capacity to model dependencies across all image tiles, which is particularly advantageous for survival prediction where global context matters. Indeed, most other models are unable to effectively handle continuous variables or survival tasks, which is a unique feature of our method and one that sets it apart from the majority of existing approaches. That said, we recognize that our study is not the culmination of this technical evolution but rather a starting point. There will always be opportunities for improvement and refinement. What our study achieves is the demonstration of a significant biomarker in a highly clinically relevant area, using a rigorously controlled cohort and introducing a unique perspective by combining spatial analysis with ctDNA—an angle that has not been previously explored. This novel integration underscores the potential of our approach in advancing the field.

2. Regarding the interpretation analysis, the study presents only a few cases with cropped patches exhibiting high attention scores, and the main conclusion that “DL can identify histopathological features linked to prognosis” is based solely on these selected patches and simple qualitative analysis. A more robust approach would involve conducting a quantitative analysis of high-attention regions across all cases in different groups. For instance, publicly available gland and nuclei segmentation models could be utilized to segment these regions, and morphological irregularity indicators (shape of glands, distribution of tumor nuclei, etc.) derived from the segmentation results could be defined and used for statistical interpretation analysis.

Response: We appreciate the reviewer’s suggestion for a more quantitative analysis of high-attention regions. While we acknowledge the value of a comprehensive quantitative approach using gland and nuclei segmentation models, our primary objective was to demonstrate the model's ability to highlight regions linked to prognostic features rather than perform an exhaustive morphological analysis. The presented heatmaps were used as illustrative examples to explore the interpretability of the model rather than as the basis for drawing statistical conclusions. However, we agree that a more quantitative assessment could provide additional insights. To address this, we have now expanded the *Limitations* section (412-415), acknowledging the potential for further quantitative investigations using segmentation tools

and morphological metrics in future studies and rephrased the Result Title to: DL as a Tool for Prognostic Histopathological Discovery.

Minor comments

1. The title “HIBRID: Histology and ct-DNA based Risk-stratification with Deep Learning” suggests a multi-modal deep learning framework (integrating histology and ctDNA) for pan-cancer patient risk stratification. However, the model presented in the manuscript is trained solely on histology images to predict risk scores, with the ctDNA component limited to plotting Kaplan-Meier (KM) curves for MRD subgroups. Additionally, the study focuses exclusively on recurrence prediction in colorectal cancer (CRC) patients, rather than a pan-cancer risk-stratification approach. I recommend that the authors revise the title to more accurately reflect the scope and subject of the study.

Response: We thank the reviewer for pointing out the need for a clearer and more specific title. To better reflect the methodology and focus of the study, we have revised the title to: "HIBRID: Histology-based Risk-stratification with Deep Learning and ctDNA in Colorectal cancer" This revised title emphasizes the primary histology-based approach while maintaining the core focus on colorectal cancer, with ctDNA analysis as a supporting element.

Reviewer #6 (Remarks to the Author): Early Career Researcher co-reviewer

- Baxter, Nancy N., Erin B. Kennedy, Emily Bergsland, Jordan Berlin, Thomas J. George, Sharlene Gill, Philip J. Gold, et al. 2022. “Adjuvant Therapy for Stage II Colon Cancer: ASCO Guideline Update.” *Journal of Clinical Oncology: Official Journal of the American Society of Clinical Oncology* 40 (8): 892–910.
- Cervantes, A., R. Adam, S. Roselló, D. Arnold, N. Normanno, J. Taïeb, J. Seligmann, et al. 2023. “Metastatic Colorectal Cancer: ESMO Clinical Practice Guideline for Diagnosis, Treatment and Follow-Up.” *Annals of Oncology : Official Journal of the European Society for Medical Oncology* 34 (1). <https://doi.org/10.1016/j.annonc.2022.10.003>.
- Chan, Gloria H. J., and Cheng E. Chee. 2019. “Making Sense of Adjuvant Chemotherapy in Colorectal Cancer.” *Journal of Gastrointestinal Oncology* 10 (6): 1183–92.
- Chau, I., and D. Cunningham. 2006. “Adjuvant Therapy in Colon Cancer--What, When and How?” *Annals of Oncology* 17 (9): 1347–59.
- Chen, Kabytto, Geoffrey Collins, Henry Wang, and James Wei Tatt Toh. 2021. “Pathological Features and Prognostication in Colorectal Cancer.” *Current Oncology (Toronto, Ont.)* 28 (6): 5356–83.
- Dosovitskiy, Alexey, Lucas Beyer, Alexander Kolesnikov, Dirk Weissenborn, Xiaohua Zhai, Thomas Unterthiner, Mostafa Dehghani, et al. 2020. “An Image Is Worth 16x16 Words: Transformers for Image Recognition at Scale.” *arXiv [cs.CV]*. arXiv. <http://arxiv.org/abs/2010.11929>.
- El Nahhas, Omar S. M., Marko van Treeck, Georg Wölflein, Michaela Unger, Marta Ligeró, Tim Lenz, Sophia J. Wagner, et al. 2025. “From Whole-Slide Image to Biomarker Prediction: End-to-End Weakly Supervised Deep Learning in Computational Pathology.” *Nature Protocols* 20 (1): 293–316.

- García-Alfonso, P., R. García-Carbonero, J. García-Foncillas, P. Pérez-Segura, R. Salazar, R. Vera, S. Ramón Y Cajal, et al. 2020. "Update of the Recommendations for the Determination of Biomarkers in Colorectal Carcinoma: National Consensus of the Spanish Society of Medical Oncology and the Spanish Society of Pathology." *Clinical & Translational Oncology: Official Publication of the Federation of Spanish Oncology Societies and of the National Cancer Institute of Mexico* 22 (11): 1976–91.
- Jiang, Xiaofeng, Michael Hoffmeister, Hermann Brenner, Hannah Sophie Muti, Tanwei Yuan, Sebastian Foersch, Nicholas P. West, et al. 2024. "End-to-End Prognostication in Colorectal Cancer by Deep Learning: A Retrospective, Multicentre Study." *The Lancet. Digital Health* 6 (1): e33–43.
- Kanemitsu, Yukihide, Yasuhiro Shimizu, Junki Mizusawa, Yoshitaka Inaba, Tetsuya Hamaguchi, Dai Shida, Masayuki Ohue, et al. 2021. "Hepatectomy Followed by mFOLFOX6 versus Hepatectomy Alone for Liver-Only Metastatic Colorectal Cancer (JCOG0603): A Phase II or III Randomized Controlled Trial." *Journal of Clinical Oncology: Official Journal of the American Society of Clinical Oncology* 39 (34): 3789–99.
- Kather, Jakob Nikolas, Lara R. Heij, Heike I. Grabsch, Chiara Loeffler, Amelie Echle, Hannah Sophie Muti, Jeremias Krause, et al. 2020. "Pan-Cancer Image-Based Detection of Clinically Actionable Genetic Alterations." *Nature Cancer* 1 (8): 789–99.
- Kiehl, Lennard, Sara Kuntz, Julia Höhn, Tanja Jutzi, Eva Krieghoff-Henning, Jakob N. Kather, Tim Holland-Letz, et al. 2021. "Deep Learning Can Predict Lymph Node Status Directly from Histology in Colorectal Cancer." *European Journal of Cancer (Oxford, England: 1990)* 157 (November):464–73.
- Koncina, Eric, Serge Haan, Stefan Rauh, and Elisabeth Letellier. 2020. "Prognostic and Predictive Molecular Biomarkers for Colorectal Cancer: Updates and Challenges." *Cancers* 12 (2): 319.
- Loeffler, Chiara Maria Lavinia, Omar S. M. El Nahhas, Hannah Sophie Muti, Zunamys I. Carrero, Tobias Seibel, Marko van Treeck, Didem Cifci, et al. 2024. "Prediction of Homologous Recombination Deficiency from Routine Histology with Attention-Based Multiple Instance Learning in Nine Different Tumor Types." *BMC Biology* 22 (1): 225.
- Luchini, C., F. Bibeau, M. J. L. Ligtenberg, N. Singh, A. Nottegar, T. Bosse, R. Miller, et al. 2019. "ESMO Recommendations on Microsatellite Instability Testing for Immunotherapy in Cancer, and Its Relationship with PD-1/PD-L1 Expression and Tumour Mutational Burden: A Systematic Review-Based Approach." *Annals of Oncology* 30 (8): 1232–43.
- Moding, Everett J., Barzin Y. Nabet, Ash A. Alizadeh, and Maximilian Diehn. 2021. "Detecting Liquid Remnants of Solid Tumors: Circulating Tumor DNA Minimal Residual Disease." *Cancer Discovery* 11 (12): 2968–86.
- Muti, Hannah Sophie, Lara Rosaline Heij, Gisela Keller, Meike Kohlruss, Rupert Langer, Bastian Dislich, Jae-Ho Cheong, et al. 2021. "Development and Validation of Deep Learning Classifiers to Detect Epstein-Barr Virus and Microsatellite Instability Status in Gastric Cancer: A Retrospective Multicentre Cohort Study." *The Lancet Digital Health*. [https://doi.org/10.1016/s2589-7500\(21\)00133-3](https://doi.org/10.1016/s2589-7500(21)00133-3).
- Muti, Hannah S., Christoph Röcken, Hans-Michael Behrens, Chiara M. L. Löffler, Nic G. Reitsam, Bianca Grosser, Bruno Märkl, et al. 2023. "Deep Learning Trained on Lymph Node Status Predicts Outcome from Gastric Cancer Histopathology: A Retrospective Multicentric Study." *European Journal of Cancer* 194 (November):113335.
- Nakamura, Yoshiaki, Jun Watanabe, Naoya Akazawa, Keiji Hirata, Kozo Kataoka, Mitsuru Yokota, Kentaro Kato, et al. 2024. "ctDNA-Based Molecular Residual Disease and Survival in Resectable Colorectal Cancer." *Nature Medicine* 30 (11): 3272–83.
- Oki, E., R. Nakanishi, K. Ando, I. Takemasa, J. Watanabe, N. Matsushashi, T. Kato, et al. 2024. "Recurrence Monitoring Using ctDNA in Patients with Metastatic Colorectal Cancer: COSMOS-CRC-03 and AURORA Studies." *ESMO Gastrointestinal Oncology* 3 (100034): 100034.
- Rebuzzi, Sara Elena, Guido Pesola, Valentino Martelli, and Alberto Felice Sobrero. 2020. "Adjuvant Chemotherapy for Stage II Colon Cancer." *Cancers* 12 (9): 2584.
- Shao, Zhuchen, Hao Bian, Yang Chen, Yifeng Wang, Jian Zhang, Xiangyang Ji, and Yongbing Zhang. 2021. "TransMIL: Transformer Based Correlated Multiple Instance Learning for Whole Slide Image Classification." *arXiv [cs.CV]*. arXiv. <http://arxiv.org/abs/2106.00908>.
- Taieb, Julien, and Claire Gallois. 2020. "Adjuvant Chemotherapy for Stage III Colon Cancer." *Cancers* 12 (9): 2679.
- Xiong, Zhi-Zhong, Ming-Hao Xie, Xian-Zhe Li, Long-Yang Jin, Feng-Xiang Zhang, Shi Yin, Hua-Xian Chen, and Lei Lian. 2023. "Risk Factors for Postoperative Recurrence in Patients with Stage II Colorectal Cancer." *BMC Cancer* 23 (1): 658.

Zhou, Jie, Ali Foroughi Pour, Hany Deirawan, Fayez Daaboul, Thazin Nwe Aung, Rafic Beydoun, Fahad Shabbir Ahmed, and Jeffrey H. Chuang. 2023. "Integrative Deep Learning Analysis Improves Colon Adenocarcinoma Patient Stratification at Risk for Mortality." *EBioMedicine* 94 (104726): 104726.

Rebuttal Letter 2

Reviewer #1 (Remarks to the Author):

Generic comments:

The authors have appropriately responded to the reviewer's comments by incorporating the necessary revisions in the Methods, especially regarding the UNI feature extraction, the subsequent transformer-based model development and training, and Results sections. The clarifications and adjustments provided contribute to a better understanding of the methodology and interpretation of the findings.

Response: We thank the reviewer for the positive feedback and appreciation of the revisions made. We are glad that the clarifications and adjustments have improved the manuscript's clarity and quality.

Specific comments:

- Substitute reference 13 by Cervantes et al for Argiles et al., that refers to localised disease.

Response: We have replaced reference 13 with the appropriate citation to Cervantes et al. and apologize for the oversight.

- L189: Add reference to figure 4 on this section.

Response: We thank the reviewer for the suggestion. We have added the reference to Figure 4A and 4B in the indicated section.

- L257: There are discrepancies between the text and the figure 2C. Which are the correct one?

Response: Well spotted. We have corrected the text to ensure consistency with Figure 2C.

- L336: I suggest including mCRC unless the benefit from adjuvant chemo remain controversial

Response: We agreed and therefore included mCRC in this part of the Discussion.

- Suppl table 1. As you have the data, I suggest including in the table the information about adjuvant chemo in the Galaxy cohort stratified by DL risk.

Response: We added the missing data into the Supplementary Table 1.

- Suppl table 2. There is a typo "adjuvant" instead of "adjuvant"

Response: We corrected the typo in the Suppl. Table 2.

- In the discussion, please emphasize the importance of your model to identify those patients that could be safely managed without ACT.

Response: We have revised the Discussion section accordingly by adding two sentences (line 359- 362) emphasizing the importance of our model in identifying patients who could potentially be safely managed without ACT.

Reviewer #2 (Remarks to the Author):

The authors have improved their manuscript considerably. However, there are still some points, which were not addressed sufficiently in the new version of the manuscript. I have outlined them below with numbers referring to my original comments.

1e+1f: Thank you to the authors for explaining to me the origin of the slides for WSIs and how synchronous tumors were handled. This explanation should also be included in the methods section.

Response: We added the explanations into the Methods section from line 118 to 119 and 126 to 128.

3: The authors refrain from including M category in their multivariable model, grouping it with other prognostic markers such as tumor budding and vascular invasion, and stating that they limit their analysis to “the clinically most relevant factors”. However, I would argue that metastatic disease is one of the MOST relevant factors for patient prognosis, as patients with stage IV disease have both higher recurrence rates and reduced survival. M category is arguably more clinically relevant than sex and age, which was included in the model by the authors. Therefore, metastatic disease should not be grouped with tumor budding and other niche histopathological risk factors. Citing another study (Jiang et al. 2024), where these factors were not included, is not a satisfactory argument, especially considering the overlap of authors between the cited study and the current manuscript. Additionally, as the authors have included stage-stratified analyses of the DL algorithm performance, the M category information is available and should be included in prognostic analyses. The authors argue that the DL model incorporates several features into a single output, but that is exactly why it is interesting if the performance of this single output is better than each of the factors individually. In their stage-stratified analysis, the authors show less prognostic power of the DL algorithm in stage IV patients. This further highlights the importance of including M category as a factor in the multivariable model.

Response: We thank the reviewer for this thoughtful and well-argued comment. In response, we have revised our multivariable analysis to include M stage, MSI status, and ACT treatment for the full GALAXY cohort (Figure 2C). As expected, due to the strong prognostic impact of M stage, the DL score was no longer an independent prognostic factor in this setting, and we have updated the corresponding results (lines 286–288) . Additionally, we included M stage in the multivariable analyses stratified by MRD status (updated Supplementary Figure 2C and 2D). These results remained consistent with our previous findings: the DL score retained prognostic significance in MRD-positive patients but not in MRD-negative patients. We agree that the M category is a major prognostic factor and appreciate the opportunity to clarify and strengthen our analysis.

7b: I appreciate the inclusion of the supplementary table outlining differences between ACT-treated and non-ACT-treated patients. However, the differences should be used in the discussion to reflect on whether these patients actually benefit from ACT treatment or the worse outcome for non-treated patients is simply because this group includes more fragile patients (higher ECOG-status, older, higher proportion of stage IV patients).

Response: We are sorry we haven't added this in the previous revision and now expanded the discussion to acknowledge that patients who did not receive ACT more frequently had adverse baseline characteristics, such as older age, higher ECOG performance status, and a greater proportion of stage IV disease (line 362-365). These differences may contribute to the worse outcomes seen in this group and introduce potential selection bias.

7c: I think the authors may have misunderstood my comment, and I apologize for the lack of clarity. The landmark analysis included to account for immortal time bias should instead have been made for the analyses regarding ACT treatment (Fig 3C-F), and not on the DL algorithm alone. It is when splitting the patients based on receiving ACT, immortal time bias may become an issue, as this is where patients have to stay alive until receiving ACT (typically 2-3 months after surgery). Patients in the ACT group are not at risk for an event until ACT has started and are therefore “immortal”.

Response: We thank the reviewer for the clarification and apologize for the misunderstanding. In response, we performed a landmark analysis with a 3-month cutoff. The stratified trends in DFS remained consistent across MRD/DL subgroups. We have added this analysis as Supplementary Figure 4 and included a corresponding note in the Results section from line 307 to 309 .

13: The authors have modified the sentence in the introduction to the following: “Despite advancements in surgical and adjuvant therapies, recurrence rates exceed 30% and 60% in resectable metastatic CRC”. I still think the wording is very confusing. It should be specified that the 30% refers to non-metastatic (I assume?) and the 60% to resectable metastatic. However, the authors should also change the “30%” figure, as this is way too high. According to the papers, the authors have cited, the recurrence rates are 12-30%, with 30% only being valid for stage III rectal cancers. Therefore, it does not “exceed 30%”.

Response: We thank the reviewer for the valuable comment. In response, we have clarified that the 30% recurrence rate refers specifically to stage III CRC and confirmed that the 60% figure pertains to resectable metastatic CRC. We have also revised the wording to more accurately reflect the reported recurrence rates see line 78

Reviewer #4 (Remarks to the Author):

Reviewer #5 (Remarks to the Author):

The authors have addressed my questions in part.

1. The conclusion here is that even though there are now established, powerful foundation models that can be used for survival modelling, a different framework was used in this study, which one could reasonably expect may have resulted in lower performance -- by how much is not known because no experiments were done. This at least requires an explanation in the limitations section.

Response: We thank the reviewer for this important observation. In response, we have added a sentence to the discussion acknowledging the limitation that recent foundation models for survival analysis were not evaluated in our study, and that direct performance comparisons are therefore not available. (line 410 to 412) and hope this satisfies the Reviewers comment.

2. The authors have not addressed this point. Given the subjectivity of this patch selection, it is not possible to state generally that the model is generally "able to highlight regions linked to prognostic features". Clearly some are, but we do not know how common this is, whether it was a coincidence, or a biased choice. The objective of providing some explainability is good, but these results are not systematic enough to support such statements. Qualitative evaluation of the patches is acceptable, but there needs to be a convincing sampling process in order to get to a solid conclusion.

Response: We thank the reviewer for this important comment. We would like to clarify that patch (tile) selection was not done subjectively, but based on a systematic approach using a Grad-CAM-like score weighted by normalized tile-level predictions. This method identifies the most influential tiles contributing to the model's output. We have clarified this process further in the Methods section.

Reviewer #6 (Remarks to the Author):

Reviewer #6 (Remarks on code availability):

I didn't run the code. But the instructions are clear and easy to follow. Should be enough to install and run the application.